# Neural Ensemble Search
# for Uncertainty Estimation and Dataset Shift

## Abstract

Ensembles of neural networks achieve superior performance compared to stand-alone networks not only in terms of predictive performance, but also uncertainty calibration and robustness to dataset shift. Diversity among networks is believed to be key for building strong ensembles, but typical approaches, such as *deep ensembles*, only ensemble different weight vectors of a fixed architecture. Instead, we propose two methods for constructing ensembles to exploit diversity among networks with *varying* architectures. We find the resulting ensembles are indeed more diverse and also exhibit better uncertainty calibration, predictive performance and robustness to dataset shift in comparison with deep ensembles on a variety of classification tasks.

## 1 Introduction

Automatically learning useful representations of data using deep neural networks has been successful across various tasks (Krizhevsky et al., 2012; Hinton et al., 2012; Mikolov et al., 2013). While some applications rely only on the predictions made by a neural network, many critical applications also require reliable predictive uncertainty estimates and robustness under the presence of dataset shift, that is, when the observed data distribution at deployment differs from the training data distribution. Examples include medical imaging (Esteva et al., 2017) and self-driving cars (Bojarski et al., 2016). However, several studies have shown that neural networks are not always robust to dataset shift (Ovadia et al., 2019; Hendrycks & Dietterich, 2019), nor do they exhibit calibrated predictive uncertainty, resulting in incorrect predictions made with high confidence (Guo et al., 2017).

Using an ensemble of networks rather than a stand-alone network improves both predictive uncertainty calibration and robustness to dataset shift. Ensembles also outperform approximate Bayesian methods (Lakshminarayanan et al., 2017; Ovadia et al., 2019; Gustafsson et al., 2020). Their success is usually attributed to the diversity among the base learners, however there are various definitions of diversity (Kuncheva & Whitaker, 2003; Zhou, 2012) without a consensus. In practice, ensembles are usually constructed by choosing a *fixed* state-of-the-art architecture and creating base learners by independently training random initializations of it. This is referred to as *deep ensembles* (Lakshminarayanan et al., 2017), a state-of-the-art method for uncertainty estimation.

However, as we show, base learners with varying network architectures make more diverse predictions. Therefore, picking a strong, fixed architecture for the ensemble's base learners neglects diversity in favor of base learner strength. This has implications for the ensemble performance, since both diversity and base learner strength are important. To overcome this, we propose *Neural Ensemble Search* (NES); a NES algorithm finds a set of diverse neural architectures that together form a strong ensemble. Note that, *a priori*, it is not obvious how to find diverse architectures that work well as an ensemble; one cannot randomly select them, since it is important to select strong ones, nor can one optimize them individually as that ignores diversity. By directly optimizing ensemble loss while maintaining independent training of base learners, a NES algorithm implicitly encourages diversity, without the need for explicitly defining diversity. In detail, our contributions are as follows:

1. We show that ensembles composed of varying architectures perform better than ensembles composed of a fixed architecture. We demonstrate that this is due to increased diversity among the ensemble's base learners (Sections 3 and 5).

2. Based on these findings and the importance of diversity, we propose two algorithms for Neural Ensemble Search: NES-RS and NES-RE. NES-RS is a simple random search based algorithm, and NES-RE is based on regularized evolution (Real et al., 2019). Both search algorithms seek performant ensembles with varying base learner architectures (Section 4).

3. With experiments on classification tasks, we evaluate the ensembles found by NES-RS and NES-RE from the point of view of both predictive performance and uncertainty calibration, comparing them to deep ensembles with fixed, optimized architectures. We find our ensembles outperform deep ensembles not only on in-distribution data but also during dataset shift (Section 5). The code for our experiments is available at: `https://anonymousfiles.io/ZaY1ccR5/`.

## 2 RELATED WORK

**Ensemble Learning and Uncertainty Estimation.** Ensembles of neural networks (Hansen & Salamon, 1990; Krogh & Vedelsby, 1995; Dietterich, 2000) are commonly used to boost performance (Szegedy et al., 2015; Simonyan & Zisserman, 2015; He et al., 2016). In practice, strategies for building ensembles include the popular approach of independently training multiple initializations of the same network (i.e. *deep ensembles* (Lakshminarayanan et al., 2017)), training base learners on bootstrap samples of the training data (i.e. *bagging*) (Zhou et al., 2002), joint training with diversity-encouraging losses (Liu & Yao, 1999; Lee et al., 2015; Zhou et al., 2018; Webb et al., 2019; Jain et al., 2020; Pearce et al., 2020) and using checkpoints during the training trajectory of a network (Huang et al., 2017; Loshchilov & Hutter, 2017). Much recent interest in ensembles has been due to their strong predictive uncertainty estimation, with extensive empirical studies (Ovadia et al., 2019; Gustafsson et al., 2020) observing that ensembles outperform other approaches for uncertainty estimation, notably including Bayesian methods (Gal & Ghahramani, 2016; Welling & Teh, 2011) and post-hoc calibration (Guo et al., 2017). Note He et al. (2020) draw a link between Bayesian methods and deep ensembles. We focus on deep ensembles as they provide state-of-the-art results in uncertainty estimation. Note that Ashukha et al. (2020) found many sophisticated ensembling techniques to be equivalent to a small-sized deep ensemble in terms of test performance.

**AutoML.** AutoML (Hutter et al., 2018) is the process of automatically designing machine learning systems. Automatic ensemble construction is commonly used in AutoML (Feurer et al., 2015). Lévesque et al. (2016) use Bayesian optimization to tune non-architectural hyperparameters of an ensemble's base learners, relying on ensemble selection (Caruana et al., 2004). Specific to neural networks, AutoML also includes neural architecture search (NAS), the process of automatically designing network architectures (Elsken et al., 2019). Existing strategies using reinforcement learning (Zoph & Le, 2017), evolutionary algorithms (Real et al., 2019) or gradient-based methods (Liu et al., 2019) have demonstrated that NAS can find architectures that surpass hand-crafted ones.

Some recent research connects ensemble learning with NAS. Methods proposed by Cortes et al. (2017) and Macko et al. (2019) iteratively add (sub-)networks to an ensemble to improve the ensemble's performance. While our work focuses on generating a diverse and well-performing (in an ensemble) set of architectures while fixing how the ensemble is built from its base learners, the aforementioned works focus on how to build the ensemble. The search spaces considered by these works are also limited compared to ours: Cortes et al. (2017) consider fully-connected layers and Macko et al. (2019) only use NASNet-A (Zoph et al., 2018) blocks with varying depth and number of filters. All aforementioned works only focus on predictive performance and do not consider uncertainty estimation and dataset shift. Concurrent to our work, Wenzel et al. (2020) consider ensembles with base learners having varying hyperparameters using an approach similar to NES-RS. However, they focus on non-architectural hyperparameters such as $L_2$ regularization strength and dropout rates, keeping the architecture fixed. As in our work, they also consider predictive uncertainty calibration and robustness to shift, finding improvements over deep ensembles.

## 3 ENSEMBLES OF VARYING ARCHITECTURES ARE MORE DIVERSE

### 3.1 DEFINITIONS AND SET-UP

Let $\mathcal{D}_{\text{train}} = \{(\boldsymbol{x}_i, y_i) : i = 1, \ldots, N\}$ be the training dataset, where the input $\boldsymbol{x}_i \in \mathbb{R}^D$ and, assuming a classification task, the output $y_i \in \{1, \ldots, C\}$. We use $\mathcal{D}_{\text{val}}$ and $\mathcal{D}_{\text{test}}$ for the validation

and test datasets, respectively. Denote by $f_\theta$ a neural network with weights $\theta$, so $f_\theta(\boldsymbol{x}) \in \mathbb{R}^C$ is the predicted probability vector over the classes for input $\boldsymbol{x}$. Let $\ell(f_\theta(\boldsymbol{x}), y)$ be the neural network's loss for data point $(\boldsymbol{x}, y)$. Given $M$ networks $f_{\theta_1}, \ldots, f_{\theta_M}$, we construct the *ensemble $F$* of these networks by averaging the outputs, yielding $F(\boldsymbol{x}) = \frac{1}{M} \sum_{i=1}^{M} f_{\theta_i}(\boldsymbol{x})$.

In addition to the ensemble's loss $\ell(F(\boldsymbol{x}), y)$, we will also consider the *average base learner* loss and the *oracle ensemble's* loss. The average base learner loss is simply defined as $\frac{1}{M} \sum_{i=1}^{M} \ell(f_{\theta_i}(\boldsymbol{x}), y)$; we use this to measure the *average base learner strength*. Similar to Lee et al. (2015); Zhou et al. (2018), the oracle ensemble $F_{\text{OE}}$ composed of base learners $f_{\theta_1}, \ldots, f_{\theta_M}$ is defined to be the function which, given an input $\boldsymbol{x}$, returns the prediction of the base learner with the smallest loss for $(\boldsymbol{x}, y)$, that is,

$$F_{\text{OE}}(\boldsymbol{x}) = f_{\theta_k}(\boldsymbol{x}), \quad \text{where} \quad k \in \operatorname*{argmin}_{i} \ell(f_{\theta_i}(\boldsymbol{x}), y).$$

Of course, the oracle ensemble can only be constructed if the true class $y$ is known. We use the oracle ensemble loss as a measure of the *diversity* in base learner predictions. Intuitively, if base learners make diverse predictions for $\boldsymbol{x}$, the oracle ensemble is more likely to find some base learner with a small loss, whereas if all base learners make identical predictions, the oracle ensemble yields the same output as any (and all) base learners. Therefore, as a rule of thumb, *small oracle ensemble loss indicates more diverse base learner predictions.*

**Proposition 3.1.** Suppose $\ell$ is negative log-likelihood (NLL). Then, the oracle ensemble loss, ensemble loss, and average base learner loss satisfy the following inequality:

$$\ell(F_{\text{OE}}(\boldsymbol{x}), y) \leq \ell(F(\boldsymbol{x}), y) \leq \frac{1}{M} \sum_{i=1}^{M} \ell(f_{\theta_i}(\boldsymbol{x}), y).$$

We refer to Appendix A for a proof. Proposition 3.1 suggests that strong ensembles require not only strong average base learners (smaller upper bound), but also more diversity in their predictions (smaller lower bound). There is extensive theoretical work relating strong base learner performance and diversity with the generalization properties of ensembles (Hansen & Salamon, 1990; Zhou, 2012; Jiang et al., 2017; Bian & Chen, 2019; Goodfellow et al., 2016). Notably, Breiman (2001) showed that the generalization error of random forests depends on the strength of individual trees and the correlation between their mistakes.

## 3.2 Visualizing Similarity in Base Learner Predictions

The fixed architecture used to build deep ensembles is typically chosen to be a strong stand-alone architecture, either hand-crafted or found by NAS. However, since ensemble performance depends not only on strong base learners but also on their diversity, optimizing the base learner's architecture and *then* constructing a deep ensemble neglects diversity in favor of strong base learner performance.

Having base learner architectures vary allows more diversity in their predictions. In this section, we provide empirical evidence for this by visualizing the base learners' predictions. Fort et al. (2019) found that base learners in a deep ensemble explore different parts of the function space by means of applying dimensionality reduction to their predictions. Building on this, we uniformly sample five architectures from the DARTS search space (Liu et al., 2019), train 20 initializations of each architecture on CIFAR-10 and visualize the similarity among the networks' predictions on the test dataset using t-SNE (Van der Maaten & Hinton, 2008). Experiment details are available in Section 5 and Appendix B.

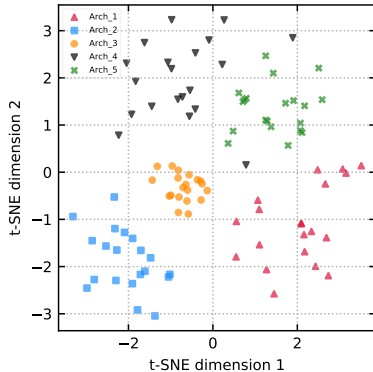

Figure 1: t-SNE visualization: five different architectures, each trained with 20 different initializations.

As shown in Figure 1, we observe clustering of predictions made by different initializations of a fixed architecture, suggesting that base learners with varying architectures explore different parts of the function space. Moreover, we also visualize the predictions of base learners of two ensembles, each

of size $M = 30$, where one is a deep ensemble and the other has varying architectures (found by NES-RS which will be introduced in Section 4). Figure 11 shows more diversity in the ensemble with varying architectures than in the deep ensemble. For each of the two ensembles shown in Figure 11, we also compute the average pairwise predictive disagreement amongst the base learners (percentage of test inputs on which two base learners disagree), which we find to be $11.88\%$ for the ensemble with varying architectures and $10.51\%$ for the ensemble with fixed architectures (this is consistent across independent runs). This indicates higher predictive diversity in the ensemble with varying architectures, in line with the t-SNE results.

## 4 Neural Ensemble Search

In this section, we introduce *neural ensemble search* (NES). In summary, a NES algorithm optimizes the architectures of base learners in an ensemble to minimize ensemble loss.

Given a network $f : \mathbb{R}^D \to \mathbb{R}^C$, let $\mathcal{L}(f, \mathcal{D}) = \sum_{(\boldsymbol{x},y) \in \mathcal{D}} \ell(f(\boldsymbol{x}), y)$ be the loss of $f$ over dataset $\mathcal{D}$. Given a set of base learners $\{f_1, \ldots, f_M\}$, let Ensemble be the function which maps $\{f_1, \ldots, f_M\}$ to the ensemble $F = \frac{1}{M} \sum_{i=1}^{M} f_i$ as defined in Section 3 . To emphasize the architecture, we use the notation $f_{\theta,\alpha}$ to denote a network with architecture $\alpha \in \mathcal{A}$ and weights $\theta$, where $\mathcal{A}$ is a search space (SS) of architectures. A NES algorithm aims to solve the following optimization problem:

$$\min_{\alpha_1,\ldots,\alpha_M \in \mathcal{A}} \mathcal{L}\left(\text{Ensemble}(f_{\theta_1,\alpha_1}, \ldots, f_{\theta_M,\alpha_M}), \mathcal{D}_{\text{val}}\right) \tag{1}$$
$$\text{s.t.} \quad \theta_i \in \operatorname*{argmin}_{\theta} \mathcal{L}(f_{\theta,\alpha_i}, \mathcal{D}_{\text{train}}) \qquad \text{for } i = 1, \ldots, M$$

Eq. 1 is difficult to solve for at least two reasons. First, we are optimizing over $M$ architectures, so the search space is effectively $\mathcal{A}^M$, compared to it being $\mathcal{A}$ in typical NAS, making it more difficult to explore fully. Second, a larger search space also increases the risk of overfitting the ensemble loss to $\mathcal{D}_{\text{val}}$. A possible approach here is to consider the ensemble as a single large network to which we apply NAS, but joint training of an ensemble through a single loss has been empirically observed to underperform training base learners independently, specially for large neural networks (Webb et al., 2019). Instead, our general approach to solve Eq. 1 consists of two steps:

1. **Pool building**: build a *pool* $\mathcal{P} = \{f_{\theta_1,\alpha_1}, \ldots, f_{\theta_K,\alpha_K}\}$ of size $K$ consisting of potential base learners, where each $f_{\theta_i,\alpha_i}$ is a network trained independently on $\mathcal{D}_{\text{train}}$.

2. **Ensemble selection**: select $M$ base learners $f_{\theta_1^*,\alpha_1^*}, \ldots, f_{\theta_M^*,\alpha_M^*}$ from $\mathcal{P}$ to form an ensemble which minimizes loss on $\mathcal{D}_{\text{val}}$. (We assume $K \geq M$.)

Step 1 reduces the options for the base learner architectures, with the intention to make the search more feasible and focus on strong architectures. Step 2 then selects a performant ensemble which implicitly encourages base learner strength *and* diversity. This procedure also ensures that the ensemble's base learners are trained independently. We propose using forward step-wise selection for step 2; that is, given the set of networks $\mathcal{P}$, we start with an empty ensemble and add to it the network from $\mathcal{P}$ which minimizes ensemble loss on $\mathcal{D}_{\text{val}}$. We repeat this without replacement until the ensemble is of size $M$. Let ForwardSelect($\mathcal{P}, \mathcal{D}_{\text{val}}, M$) denote the set of $M$ base learners selected from $\mathcal{P}$ by this procedure.

Note that selecting the ensemble from $\mathcal{P}$ is a combinatorial optimization problem; a greedy approach such as ForwardSelect is nevertheless effective (Caruana et al., 2004), while keeping computational overhead low, given the predictions of the networks on $\mathcal{D}_{\text{val}}$. We also experimented with three other ensemble selection algorithms: (1) Starting with the best network by validation performance, add the next best network to the ensemble only if it improves validation performance, iterating until the ensemble size is $M$ or all models have been considered.[1] (2) Select the top $M$ networks by validation performance. (3) Forward step-wise selection *with* replacement. We typically found that these three performed comparatively or worse than our choice ForwardSelect.

We have not yet discussed the algorithm for building the pool in step 1; we propose two approaches, NES-RS (Section 4.1) and NES-RE (Section 4.2). NES-RS is a simple random search based algorithm, while NES-RE is based on regularized evolution (Real et al., 2019), a state-of-the-art NAS algorithm.

---

[1]This approach returns an ensemble of size at most $M$.

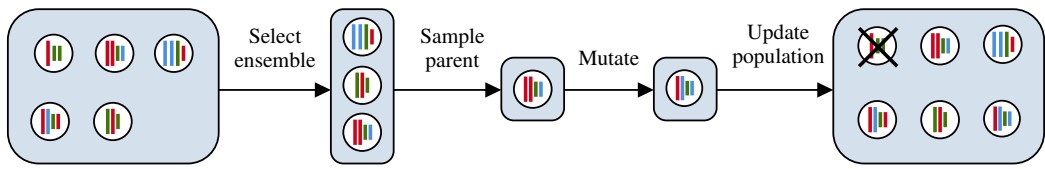

Figure 2: Illustration of one iteration of NES-RE. Network architectures are represented as colored bars of different lengths illustrating different layers and widths. Starting with the current population, ensemble selection is applied to select parent candidates, among which one is sampled as the parent. A mutated copy of the parent is added to the population, and the oldest member is removed.

Note that while gradient-based NAS methods have recently become popular, they are not naively applicable in our setting as the base learner selection component `ForwardSelect` is typically non-differentiable.

## 4.1 NES with Random Search

In NAS, random search (RS) is a competitive baseline on carefully designed architecture search spaces (Li & Talwalkar, 2019; Yang et al., 2020; Yu et al., 2020). Motivated by its success and simplicity, we first introduce NES with random search (NES-RS). NES-RS builds the pool $\mathcal{P}$ by independently sampling architectures uniformly from the search space $\mathcal{A}$ (and training them). Since the architectures of networks in $\mathcal{P}$ vary, applying ensemble selection is a simple way to exploit diversity, yielding a performant ensemble. Algorithm 1 describes NES-RS in pseudocode.

---

**Algorithm 1:** NES with Random Search

**Data:** Search space $\mathcal{A}$; ensemble size $M$; comp. budget $K$; $\mathcal{D}_{\text{train}}, \mathcal{D}_{\text{val}}$.
1 Sample $K$ architectures $\alpha_1, \ldots, \alpha_K$ independently and uniformly from $\mathcal{A}$.
2 Train each architecture $\alpha_i$ using $\mathcal{D}_{\text{train}}$, yielding a pool of networks $\mathcal{P} = \{f_{\theta_1, \alpha_1}, \ldots, f_{\theta_K, \alpha_K}\}$.
3 Select base learners $\{f_{\theta_1^*, \alpha_1^*}, \ldots, f_{\theta_M^*, \alpha_M^*}\} = \texttt{ForwardSelect}(\mathcal{P}, \mathcal{D}_{\text{val}}, M)$ by forward step-wise selection without replacement.
4 **return** ensemble $\texttt{Ensemble}(f_{\theta_1^*, \alpha_1^*}, \ldots, f_{\theta_M^*, \alpha_M^*})$

---

## 4.2 NES with Regularized Evolution

A more guided approach for building the pool $\mathcal{P}$ is using regularized evolution (RE) (Real et al., 2019). While RS has the benefit of simplicity, by sampling architectures uniformly, the resulting pool might contain many weak architectures, leaving few strong architectures for `ForwardSelect` to choose between. Therefore, NES-RS might require a large pool in order to explore interesting parts of the search space. RE is an evolutionary algorithm used for NAS which explores the search space by evolving a *population* of architectures. In summary, RE starts with a randomly initialized fixed-size population of architectures. At each iteration, a subset of size $m$ of the population is sampled, from which the best network by validation loss is selected as the parent. A mutated copy of the parent architecture, called the child, is trained and added to the population, and the oldest member of the population is removed, preserving the population size. This is iterated until the computational budget is reached, returning the *history*, i.e. all the networks evaluated during the search, from which the best model is chosen by validation loss.

Based on RE for NAS, we propose NES-RE to build the pool of potential base learners. NES-RE starts by randomly initializing a population $\mathfrak{p}$ of size $P$. At each iteration, we apply `ForwardSelect` to the population to select an ensemble of size $m$, and we uniformly sample one base learner from the ensemble to be the parent. A mutated copy of the parent is added to $\mathfrak{p}$ and the oldest network is removed, as in regularized evolution. This process is repeated until the computational budget is reached, and the history is returned as the pool $\mathcal{P}$. See Algorithm 2 for pseudocode and Figure 2 for an illustration.

Also, note the distinction between the *population* and the *pool* in NES-RE: the population is evolved, whereas the pool is the set of all networks evaluated during evolution (i.e., the history) and is used

---

**Algorithm 2:** NES with Regularized Evolution

**Data:** Search space $\mathcal{A}$; ensemble size $M$; comp. budget $K$; $\mathcal{D}_{\text{train}}, \mathcal{D}_{\text{val}}$; population size $P$;
       number of parent candidates $m$.

1   Sample $P$ architectures $\alpha_1, \ldots, \alpha_P$ independently and uniformly from $\mathcal{A}$.

2   Train each architecture $\alpha_i$ using $\mathcal{D}_{\text{train}}$, and initialize $\mathfrak{p} = \mathcal{P} = \{f_{\theta_1, \alpha_1}, \ldots, f_{\theta_P, \alpha_P}\}$.

3   **while** $|\mathcal{P}| < K$ **do**

4      Select $m$ parent candidates $\{f_{\widetilde{\theta}_1, \widetilde{\alpha}_1}, \ldots, f_{\widetilde{\theta}_m, \widetilde{\alpha}_m}\} = \texttt{ForwardSelect}(\mathfrak{p}, \mathcal{D}_{\text{val}}, m)$.

5      Sample uniformly a parent architecture $\alpha$ from $\{\widetilde{\alpha}_1, \ldots, \widetilde{\alpha}_m\}$.     `// α stays in 𝔭.`

6      Apply mutation to $\alpha$, yielding child architecture $\beta$.

7      Train $\beta$ using $\mathcal{D}_{\text{train}}$ and add the trained network $f_{\theta, \beta}$ to $\mathfrak{p}$ and $\mathcal{P}$.

8      Remove the oldest member in $\mathfrak{p}$.     `// as done in RE (Real et al., 2019).`

9   Select base learners $\{f_{\theta_1^*, \alpha_1^*}, \ldots, f_{\theta_M^*, \alpha_M^*}\} = \texttt{ForwardSelect}(\mathcal{P}, \mathcal{D}_{\text{val}}, M)$ by forward
   step-wise selection without replacement.

10 **return** ensemble $\texttt{Ensemble}(f_{\theta_1^*, \alpha_1^*}, \ldots, f_{\theta_M^*, \alpha_M^*})$

---

post-hoc for selecting the ensemble, similar to Real et al. (2019). Moreover, `ForwardSelect` is used both for selecting $m$ parent candidates (line 4 in NES-RE) and choosing the final ensemble of size $M$ (line 9 in NES-RE). In general, $m \neq M$.

### 4.3   ENSEMBLE ADAPTATION TO DATASET SHIFT

Using deep ensembles is a common way of building a model robust to distributional shift relative to training data. In general, one may not know the type of distributional shift that occurs at test time. However, by using an ensemble, diversity in base learner predictions prevents the model from relying on one base learner's predictions which may not only be incorrect but also overconfident.

We assume that one does not have access to data points with test-time shift at training time, but one does have access to some validation data $\mathcal{D}_{\text{val}}^{\text{shift}}$ with a *validation* shift, which encapsulates one's belief about test-time shift. Crucially, test and validation shifts are disjoint. A simple way to adapt NES-RS and NES-RE to return ensembles robust to shift is by using $\mathcal{D}_{\text{val}}^{\text{shift}}$ instead of $\mathcal{D}_{\text{val}}$ whenever applying `ForwardSelect` to select the final ensemble. In algorithms 1 and 2, this is in lines 3 and 9, respectively. Note that in line 4 of Algorithm 2, we can also replace $\mathcal{D}_{\text{val}}$ with $\mathcal{D}_{\text{val}}^{\text{shift}}$ when expecting test-time shift, however to avoid running NES-RE once for each of $\mathcal{D}_{\text{val}}, \mathcal{D}_{\text{val}}^{\text{shift}}$, we simply sample one of $\mathcal{D}_{\text{val}}, \mathcal{D}_{\text{val}}^{\text{shift}}$ uniformly at each iteration, in order to explore architectures that work well both in-distribution and during shift. See Appendices C.1.3 and B.4 for further discussion.

## 5   EXPERIMENTS

We compare NES to deep ensembles on different choices of architecture search space (DARTS and NAS-Bench-201 (Dong & Yang, 2020) search spaces) and dataset (FMNIST, CIFAR-10, CIFAR-100, ImageNet-16-120 and Tiny ImageNet). For CIFAR-10/100 and Tiny ImageNet, we also consider dataset shifts proposed by Hendrycks & Dietterich (2019). The metrics used are: NLL, classification error and expected calibration error (ECE) (Guo et al., 2017; Naeini et al., 2015). Hyperparame-

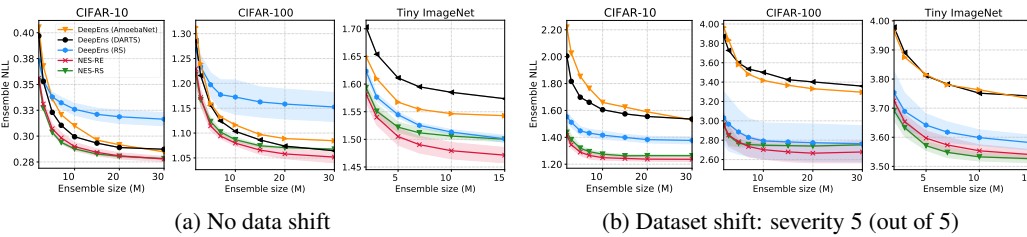

(a) No data shift          (b) Dataset shift: severity 5 (out of 5)

Figure 3: NLL vs. ensemble sizes on CIFAR-10, CIFAR-100 and Tiny ImageNet with and without respective dataset shifts (Hendrycks & Dietterich, 2019) over DARTS search space.

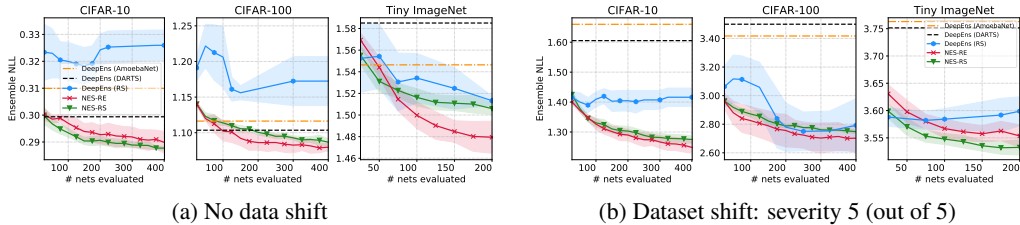

(a) No data shift          (b) Dataset shift: severity 5 (out of 5)

Figure 4: NLL vs. budget $K$ on CIFAR-10/100 and Tiny ImageNet with and without respective dataset shifts over the DARTS search space. Ensemble size is fixed at $M = 10$.

ter choices, experimental and implementation details are available in Appendix B. Unless stated otherwise, all evaluations are on the test dataset. Additional experimental results are in Appendix C.

**Baselines.** We compare the ensembles found by NES to the baseline of deep ensembles built using a fixed, optimized architecture. On the DARTS search space, the fixed architecture is either: (1) optimized by random search, called DeepEns (RS), (2) the architecture found using the DARTS algorithm, called DeepEns (DARTS) or (3) the architecture found using RE, called DeepEns (AmoebaNet). On the NAS-Bench-201 search space, instead of DeepEns (DARTS/AmoebaNet), we compare to DeepEns (GDAS), found using GDAS (Dong & Yang, 2019)[2], and DeepEns (Optimal), which uses the optimal architecture in the NAS-Bench-201 search space by validation loss. All base learners use the same training routine and have approximately the same model size for a given search space and dataset. See Appendix B for details and further discussion.

**Computational cost analysis.** NES algorithms typically have a lower computational cost than deep ensembles of optimized architectures. There are two sources of computational cost for the deep ensemble baselines: 1. searching for the fixed base learner architecture (which involves training and evaluating some number of architectures depending on the NAS algorithm used) and 2. training $M$ initializations of it. On the other hand, NES does not require these two steps; the only source of cost for NES-RS and NES-RE is training the networks to build a pool, because ensemble selection from a trained pool of networks is much cheaper. Running a NES algorithm with pool size $K$ requires training only $K$ networks. However, DeepEns (RS), for instance, has cost $k + M$ where $k$ is the number of networks trained during random search to find the optimized architecture. In our experiments, we set $k = K$, so NES-RE and NES-RS are cheaper than DeepEns (RS). Figure 4 compares NES algorithms and DeepEns (RS) as a function of $K$. Appendix C.3 describes a study in which

Table 1: Classification error of ensembles for different shift severities. Best values and all values within $95\%$ confidence interval are bold faced.

(a) $M = 10$; DARTS search space.

| Dataset | Shift Severity | Classif. error (%), $\mathcal{A}$ = DARTS search space | | | | |
|---|---|---|---|---|---|---|
| | | DeepEns (Amoe.) | DeepEns (DARTS) | DeepEns (RS) | NES-RS | NES-RE |
| C10 | 0 | 9.7 | 10.0 | $10.8_{\pm0.2}$ | $\mathbf{9.4}_{\pm0.1}$ | $\mathbf{9.4}_{\pm0.2}$ |
| | 3 | 25.6 | 26.3 | $25.1_{\pm0.7}$ | $23.2_{\pm0.2}$ | $\mathbf{22.9}_{\pm0.2}$ |
| | 5 | 42.7 | 42.9 | $41.1_{\pm0.9}$ | $38.0_{\pm0.2}$ | $\mathbf{37.4}_{\pm0.4}$ |
| C100 | 0 | 31.6 | 30.9 | $33.2_{\pm1.2}$ | $\mathbf{30.7}_{\pm0.1}$ | $\mathbf{30.4}_{\pm0.4}$ |
| | 3 | 54.2 | 55.1 | $54.8_{\pm1.6}$ | $\mathbf{49.8}_{\pm0.1}$ | $\mathbf{49.1}_{\pm1.0}$ |
| | 5 | 68.5 | 68.5 | $64.3_{\pm3.2}$ | $\mathbf{62.4}_{\pm0.2}$ | $\mathbf{61.4}_{\pm1.4}$ |
| Tiny ImageNet | 0 | 38.5 | 39.1 | $\mathbf{37.5}_{\pm0.3}$ | $\mathbf{37.4}_{\pm0.2}$ | $\mathbf{37.0}_{\pm0.6}$ |
| | 3 | 55.4 | 54.6 | $53.8_{\pm0.6}$ | $53.0_{\pm0.2}$ | $\mathbf{52.7}_{\pm0.2}$ |
| | 5 | 71.7 | 71.6 | $70.5_{\pm0.4}$ | $\mathbf{69.9}_{\pm0.2}$ | $70.2_{\pm0.1}$ |

(b) $M = 3$; NAS-Bench-201 search space.

| Dataset | Shift Severity | Classif. error (%), $\mathcal{A}$ = NAS-Bench-201 search space | | | | |
|---|---|---|---|---|---|---|
| | | DeepEns (GDAS) | DeepEns (Optimal) | DeepEns (RS) | NES-RS | NES-RE |
| C10 | 0 | 8.4 | $\mathbf{7.2}$ | $7.8_{\pm0.2}$ | $7.7_{\pm0.1}$ | $7.6_{\pm0.1}$ |
| | 3 | 28.7 | 27.1 | $28.3_{\pm0.3}$ | $\mathbf{22.0}_{\pm0.2}$ | $22.5_{\pm0.1}$ |
| | 5 | 47.8 | 46.3 | 37.1 | $\mathbf{32.5}_{\pm0.2}$ | $33.0_{\pm0.5}$ |
| C100 | 0 | 29.9 | 26.4 | $26.3_{\pm0.4}$ | $\mathbf{23.3}_{\pm0.3}$ | $23.5_{\pm0.2}$ |
| | 3 | 60.3 | 54.5 | $57.0_{\pm0.9}$ | $\mathbf{46.6}_{\pm0.3}$ | $46.7_{\pm0.5}$ |
| | 5 | 75.3 | 69.9 | $64.5_{\pm0.0}$ | $\mathbf{59.7}_{\pm0.2}$ | $60.0_{\pm0.6}$ |
| ImageNet-16-120 | 0 | 49.9 | 49.9 | $50.5_{\pm0.6}$ | $48.1_{\pm1.0}$ | $\mathbf{47.9}_{\pm0.4}$ |

we consider another (more expensive) baseline: optimize the fixed architecture, train $K$ initializations of it and use `ForwardSelect` to select an ensemble of size $M$ (where $M < K$). Appendix C.3 also includes further discussion of cost.

**NLL on in-distribution and shifted data.** Figures 3a and 4a show the NLL achieved by NES-RS, NES-RE and the baselines as functions of the ensemble size $M$ and budget $K$ respectively for in-distribution data. We find that NES algorithms consistently outperform deep ensembles across

---

[2]We did not consider DARTS algorithm on NAS-Bench-201, since it returns degenerate architectures with poor performance on this space (Dong & Yang, 2020). GDAS, on the other hand, yields state-of-the-art performance on this space (Dong & Yang, 2020).

ensemble sizes, with NES-RE usually outperforming NES-RS. Next, we evaluate the robustness of the ensembles to dataset shift in Figures 3b and 4b. All base learners are trained on $\mathcal{D}_{\text{train}}$ without data augmentation with shifted examples. However, we use a shifted validation dataset, $\mathcal{D}_{\text{val}}^{\text{shift}}$, and a shifted test dataset, $\mathcal{D}_{\text{test}}^{\text{shift}}$. $\mathcal{D}_{\text{val}}^{\text{shift}}$ is built by applying a random *validation shift* to each datapoint in $\mathcal{D}_{\text{val}}$. $\mathcal{D}_{\text{test}}^{\text{shift}}$ is built similarly but using instead a disjoint set of *test shifts* applied to $\mathcal{D}_{\text{test}}$; see Appendix B and Hendrycks & Dietterich (2019) for details. The severity of the shift varies between 1-5. The fixed architecture used in the baseline DeepEns (RS) is selected based on its loss over $\mathcal{D}_{\text{val}}^{\text{shift}}$, but the DARTS and AmoebaNet architectures remain unchanged. As shown in Figures 3b and 4b, ensembles picked by NES-RS and NES-RE are more robust to dataset shift than all three baselines. Unsurprisingly, DeepEns (DARTS/AmoebaNet) perform poorly compared to the other methods, as they are not optimized to deal with dataset shift, highlighting that highly optimized architectures can fail heavily under dataset shift.

**Classification error and uncertainty calibration.** We also assess the ensembles using classification error and expected calibration error (ECE). ECE measures the mismatch between the model's confidence and the corresponding achieved accuracy at different levels of confidence. As shown in Figure 5, ensembles found with NES tend to exhibit superior uncertainty calibration and are either competitive with or outperform deep ensembles for most shift severities. Notably, on CIFAR-10, ECE reduces by

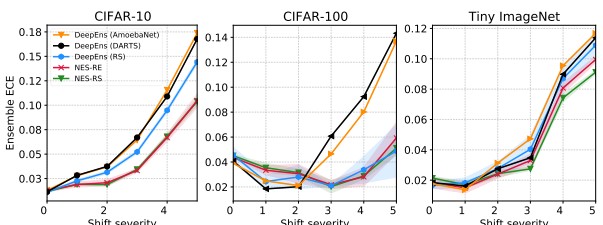

Figure 5: ECE vs. dataset shift severity on CIFAR-10, CIFAR-100 and Tiny ImageNet over the DARTS search space. No dataset shift is indicated as severity 0. Ensemble size is fixed at $M = 10$.

up to $40\%$ compared to deep ensembles. Note that good uncertainty calibration is especially important when models are used during dataset shift. In terms of classification error, we find that ensembles built by NES consistently outperform deep ensembles, with reductions of up to 7 percentage points in error, shown in Table 1. As with NLL, NES-RE tends to outperforms NES-RS.

**Diversity and average base learner strength.** To understand why ensembles found by NES algorithms outperform deep ensembles with fixed, optimized architectures, we view the ensembles through the lenses of the average base learner loss and oracle ensemble loss as defined in Section 3 and shown in Figure 6. Recall that *small oracle ensemble loss indicates higher diversity*. We see that NES finds ensembles with smaller oracle ensemble losses indicating greater diversity among base learners. Unsurprisingly, the average base learner is occasionally weaker for NES as compared to DeepEns (RS). Despite this, the ensemble performs better (Figure 3), highlighting once again the importance of diversity.

**Results on the NAS-Bench-201 search space.** We also compare NES to deep ensembles for the NAS-Bench-201 search space, which has two benefits: this shows our findings are not specific to the DARTS search space, and NAS-Bench-201 is a search space for which all architectures' trained weights are available, which allows us to compare NES to the deep ensemble of the *optimal* architecture by validation loss. Results shown in Figure 7 compare the losses of the en-

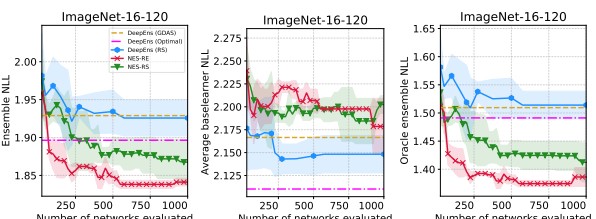

Figure 7: Ensemble, average base learner and oracle ensemble NLL for ImageNet-16-120 on the NAS-Bench-201 search space. Ensemble size $M = 3$.

semble, average base learner and oracle ensemble. Interestingly, despite DeepEns (Optimal) having a significantly lower average base learner loss than other methods, the lack of diversity (as indicated by higher oracle ensemble loss) yields an ensemble which is outperformed by both NES algorithms.

**Further comparisons and studies.** In Appendices C.3 and C.4, we compare NES to a number of additional baselines. In Appendix C.3, we consider deep ensembles with ensemble selection over

Figure 6: Average base learner loss and oracle ensemble loss (see Section 3 for definitions) on CIFAR-10/100 and Tiny ImageNet. Recall that small oracle ensemble loss generally corresponds to higher diversity. These findings are qualitatively consistent across datasets and also over shifted test data. See Appendix C.1.

random initializations of a fixed architecture to ascertain whether the improvement in NES is only due to ensemble selection. Our results show that NES continues to outperform this baseline affirming the importance of varying the architecture. In Appendix C.4, we consider baselines of ensembles with other hyperparameters being varied. In particular, we consider ensembles with a fixed, optimized architecture but varying learning rates and $L_2$ regularization strengths (similar to concurrent work by Wenzel et al. (2020)) and ensembles with architectures varying only in terms of width and depth. The results again show that NES tends to improve upon these baselines.

## 6 CONCLUSION

We showed that ensembles with varying architectures are more diverse than ensembles with fixed architectures and argued that deep ensembles with fixed, optimized architectures neglect diversity. To this end, we proposed *Neural Ensemble Search*, which exploits diversity between base learners of varying architectures to find strong ensembles. We demonstrated empirically that NES-RE and NES-RS outperform deep ensembles in terms of both predictive performance and uncertainty calibration, for in-distribution data and also during dataset shift. We found that even NES-RS, a simple random search based algorithm, found ensembles capable of outperforming deep ensembles built with state-of-the-art architectures.

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

# A PROOF OF PROPOSITION 3.1

Taking the loss function to be NLL, we have $\ell(f(\boldsymbol{x}), y)) = -\log[f(\boldsymbol{x})]_y$, where $[f(\boldsymbol{x})]_y$ is the probability assigned by the network $f$ of $\boldsymbol{x}$ belonging to the true class $y$, i.e. indexing the predicted probabilities $f(\boldsymbol{x})$ with the true target $y$. Note that $t \mapsto -\log t$ is a convex and decreasing function.

We first prove $\ell(F_{\text{OE}}(\boldsymbol{x}), y) \le \ell(F(\boldsymbol{x}), y)$. Recall, by definition of $F_{\text{OE}}$, we have $F_{\text{OE}}(\boldsymbol{x}) = f_{\theta_k}(\boldsymbol{x})$ where $k \in \operatorname{argmin}_i \ell(f_{\theta_i}(\boldsymbol{x}), y)$, therefore $[F_{\text{OE}}(\boldsymbol{x})]_y = [f_{\theta_k}(\boldsymbol{x})]_y \ge [f_{\theta_i}(\boldsymbol{x})]_y$ for all $i = 1, \ldots, C$. That is, $f_{\theta_k}$ assigns the highest probability to the correct class $y$ for input $\boldsymbol{x}$. Since $-\log$ is a decreasing function, we have

$$\ell(F(\boldsymbol{x}), y) = -\log\left(\frac{1}{M}\sum_{i=1}^{M}[f_{\theta_i}(\boldsymbol{x})]_y\right) \ge -\log\left([f_{\theta_k}(\boldsymbol{x})]_y\right) = \ell(F_{\text{OE}}(\boldsymbol{x}), y).$$

We apply Jensen's inequality in its finite form for the second inequality. Jensen's inequality states that for a real-valued, convex function $\varphi$ with its domain being a subset of $\mathbb{R}$ and numbers $t_1, \ldots, t_n$ in its domain, $\varphi(\frac{1}{n}\sum_{i=1}^{n}t_i) \le \frac{1}{n}\sum_{i=1}^{n}\varphi(t_i)$. Noting that $-\log$ is a convex function, $\ell(F(\boldsymbol{x}), y) \le \frac{1}{M}\sum_{i=1}^{M}\ell(f_{\theta_i}(\boldsymbol{x}), y)$ follows directly.

# B EXPERIMENTAL AND IMPLEMENTATION DETAILS

We describe details of the experiments shown in Section 5 and Appendix C. Note that unless stated otherwise, all sampling over a discrete set is done uniformly in the discussion below.

## B.1 ARCHITECTURE SEARCH SPACES

**DARTS search space.** The first architecture search space we consider in our experiments is the one from DARTS (Liu et al., 2019). We search for two types of *cells*: *normal* cells, which preserve the spatial dimensions, and *reduction* cells, which reduce the spatial dimensions. These cells are stacked using a pre-determined macro-architecture where they are usually repeated and connected using additional skip connections. Each cell is a directed acyclic graph, where nodes represent feature maps in the computational graph and edges between them correspond to operation choices (e.g. a convolution operation). The cell parses inputs from the previous and previous-previous cells in its 2 input nodes. Afterwards it contains 5 nodes: 4 intermediate nodes that aggregate the information coming from 2 previous nodes in the cell and finally an output node that concatenates the output of all intermediate nodes across the channel dimension. AmoebaNet contains one more intermediate node, making that a deeper architecture. The set of possible operations (eight in total in DARTS) that we use for each edge in the cells is the same as DARTS, but we leave out the "zero" operation since that is not necessary for non-differentiable approaches such as random search and evolution. Randomly of architectures is done by sampling the structure of the cell and the operations at each edge. The total number of architectures contained in this space is $\approx 10^{18}$. We refer the reader to Liu et al. (2019) for more details.

**NAS-Bench-201 search space.** NAS-Bench-201 (Dong & Yang, 2020) is a tabular NAS benchmark, i.e. all architectures in the cell search space are trained and evaluated beforehand so one can query their performance (and weights) from a table quickly. Since this space is exhaustively evaluated, its size is also limited to only *normal* cells containing 4 nodes in total (1 input, 2 intermediate and 1 output node) and 5 operation choices on every edge connecting two nodes. This means that there are only 15,625 possible architecture configurations in this space. The networks are constructed by stacking 5 cells with in-between fixed residual blocks for reducing the spacial resolution. Each of them is trained for 200 epochs 3 times with 3 different seeds on 3 image classification datasets. For more details, please refer to Dong & Yang (2020).

## B.2 DATASETS

**Fashion-MNIST (Xiao et al., 2017).** Fashion-MNIST consists of a training set of 60k 28×28 grayscale images and a test set of 10k images. The number of total labels is 10 classes. We split the 60k training set images to 50k used to train the networks and 10k used only for validation.

**CIFAR-10/100 (Krizhevsky et al., 2009).** CIFAR-10 and CIFAR-100 both consist of 60k 32×32 colour images with 10 and 100 classes, respectively. We use 10k of the 60k training images as the validation set. We use the 10k original test set for final evaluation.

**Tiny ImageNet (Le & Yang, 2015).** Tiny Imagenet has 200 classes and each class has 500 training, 50 validation and 50 test colour images with 64×64 resolution. Since the original test labels are not available, we split the 10k validation examples into 5k for testing and 5k for validation.

**ImageNet-16-120 (Dong & Yang, 2020)** This variant of the ImageNet-16-120 (Chrabaszcz et al., 2017) contains 151.7k train, 3k validation and 3k test ImageNet images downsampled to 16×16 and 120 classes.

Note that the test data points are only used for final evaluation. The data points for validation are used by the NES algorithms during ensemble selection and by DeepEns (RS) for picking the best architecture from the pool to use in the deep ensemble. Note that when considering dataset shift for CIFAR-10, CIFAR-100 and Tiny ImageNet, we also apply two disjoint sets of "corruptions" (following the terminology used by (Hendrycks & Dietterich, 2019)) to the validation and test sets. We never apply any corruption to the training data. More specifically, out of the 19 different corruptions provided by Hendrycks & Dietterich (2019), we randomly apply one from {`Speckle Noise`, `Gaussian Blur`, `Spatter`, `Saturate`} to each data point in the validation set and one from {`Gaussian Noise`, `Shot Noise`, `Impulse Noise`, `Defocus Blur`, `Glass Blur`, `Motion Blur`, `Zoom Blur`, `Snow`, `Frost`, `Fog`, `Brightness`, `Contrast`, `Elastic Transform`, `Pixelate`, `JPEG compression`} to each data point in the test set. This choice of validation and test corruptions follows the recommendation of (Hendrycks & Dietterich, 2019). Also, as mentioned in Section 5, each of these corruptions has 5 severity levels, which yields 5 corresponding severity levels for $\mathcal{D}_{\text{val}}^{\text{shift}}$ and $\mathcal{D}_{\text{test}}^{\text{shift}}$.

### B.3 Training Routine

The macro-architecture we use has 16 initial channels and 8 cells (6 normal and 2 reduction), and was trained using a batch size of 100 for 100 epochs for CIFAR-10 and CIFAR-100 and 15 epochs for Fashion-MNIST. For Tiny ImageNet, we used a batch size of 128 for 100 epochs. Unlike DARTS, we do not use any data augmentation procedure during training, nor any additional regularization such as ScheduledDropPath (Zoph et al., 2018) or auxiliary heads, except for the case of Tiny ImageNet, for which we used ScheduledDropPath and standard data augmentation as default in DARTS. All other hyperparameter settings are exactly as in DARTS (Liu et al., 2019).

All results shown are averaged over multiple runs with error bars indicating a 95% confidence interval. We used a budget $K = 400$ in all experiments, except Tiny ImageNet on the DARTS search space, which used $K = 200$ and ImageNet-16-120 on the NAS-Bench-201 search space, which used $K = 1000$.

### B.4 Implementation Details of NES-RE

**Parallization.** Running NES-RE on a single GPU requires evaluating hundreds of networks sequentially, which is tedious. To circumvent this, we distribute the "while $|\mathcal{P}| < K$" loop in Algorithm 2 over multiple GPUs, called worker nodes. We use the parallelism scheme provided by the `hpbandster` (Falkner et al., 2018) codebase.[3] In brief, the master node keeps track of the population and history (lines 1, 4-6, 8 in Algorithm 2), and it distributes the training of the networks to the individual worker nodes (lines 2, 7 in Algorithm 2). In our experiments, we always use 20 worker nodes and evolve a population $\mathfrak{p}$ of size $P = 50$ when working over the DARTS search space. Over NAS-Bench-201, we used one worker since it is a tabular NAS benchmark and hence is quick to evaluate on. During iterations of evolution, we use an ensemble size of $m = 10$ to select parent candidates.

**Mutations.** We adapt the mutations used in RE to the DARTS search space. As in RE, we first pick a normal or reduction cell at random to mutate and then sample one of the following mutations:

---

[3] `https://github.com/automl/HpBandSter`

- `identity`: no mutation is applied to the cell.
- `op mutation`: sample one edge in the cell and replace its operation with another operation sampled from the list of operations.
- `hidden state mutation`: sample one intermediate node in the cell, then sample one of its two incoming edges. Replace the input node of that edge with another sampled node, without altering the edge's operation.

See Real et al. (2019) for details and illustrations of these mutations. Note that for NAS-Bench-201, following Dong & Yang (2020) we only use `op mutation`.

**Adaptation of NES-RE to dataset shifts.** As described in Section 4.3, at each iteration of evolution, the validation set used in line 4 of Algorithm 2 is sampled uniformly between $\mathcal{D}_{\text{val}}$ and $\mathcal{D}_{\text{val}}^{\text{shift}}$ when dealing with dataset shift. In this case, we use shift severity level 5 for $\mathcal{D}_{\text{val}}^{\text{shift}}$. Once the evolution is complete and the pool $\mathcal{P}$ has been formed, then for each severity level $s \in \{0, 1, \ldots, 5\}$, we apply `ForwardSelect` with $\mathcal{D}_{\text{val}}^{\text{shift}}$ of severity $s$ to select an ensemble from $\mathcal{P}$ (line 9 in Algorithm 2), which is then evaluated on $\mathcal{D}_{\text{test}}^{\text{shift}}$ of severity $s$. (Here $s = 0$ corresponds to no shift.) This only applies to CIFAR-10, CIFAR-100 and Tiny ImageNet, as we do not consider dataset shift for Fashion-MNIST and ImageNet-16-120 .

## C ADDITIONAL EXPERIMENTS

In this section we provide additional results for the experiments conducted in Section 5. Note that, as with all results shown in Section 5, all evaluations are made on test data unless stated otherwise.

### C.1 ADDITIONAL RESULTS ON THE DARTS SEARCH SPACE

#### C.1.1 RESULTS ON FASHION-MNIST

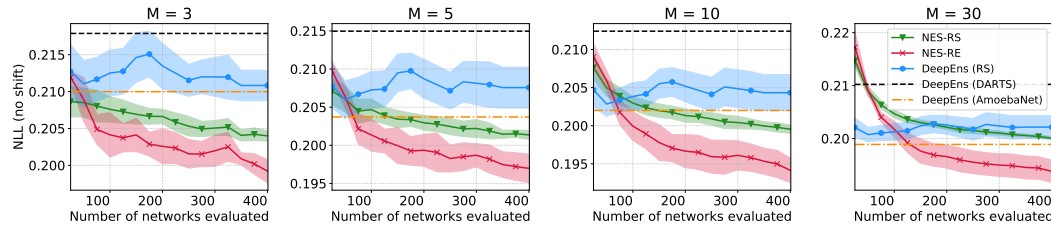

Figure 7: Results on Fashion-MNIST with varying ensembles sizes $M$. Lines show the mean NLL achieved by the ensembles with 95% confidence intervals.

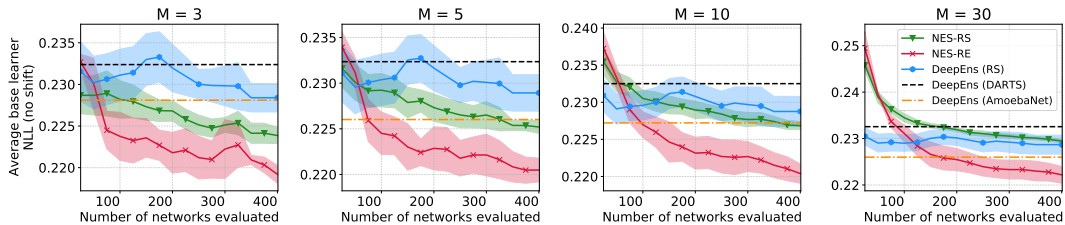

Figure 8: Average base learner loss for NES-RS, NES-RE and DeepEns (RS) on Fashion-MNIST. Lines show the mean NLL and 95% confidence intervals.

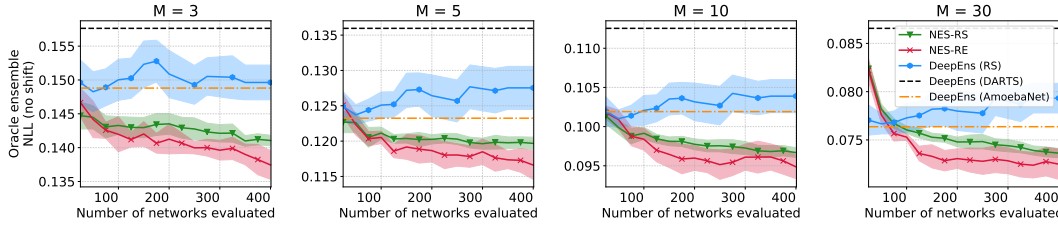

Figure 9: Oracle ensemble loss for NES-RS, NES-RE and DeepEns (RS) on Fashion-MNIST. Lines show the mean NLL and 95% confidence intervals.

As shown in Figure 7, we see a similar trend on Fashion-MNIST as with other datasets: NES ensembles outperform deep ensembles with NES-RE outperforming NES-RS. To understand why NES algorithms outperform deep ensembles on Fashion-MNIST (Xiao et al., 2017), we compare the average base learner loss (Figure 8) and oracle ensemble loss (Figure 9) of NES-RS, NES-RE and DeepEns (RS). Notice that, apart from the case when ensemble size $M = 30$, NES-RS and NES-RE find ensembles with both stronger and more diverse base learners (smaller losses in Figures 8 and 9, respectively). While it is expected that the oracle ensemble loss is smaller for NES-RS and NES-RE compared to DeepEns (RS), it initially appears surprising that DeepEns (RS) has a larger average base learner loss considering that the architecture for the deep ensemble is chosen to minimize the base learner loss. We found that this is due to the loss having a sensitive dependence not only on the architecture but also the initialization of the base learner networks. Therefore, re-training the best architecture by validation loss to build the deep ensemble yields base learners with higher losses due to the use of different random initializations. Fortunately, NES algorithms are not affected by this, since they simply select the ensemble's base learners from the pool without having to re-train anything

which allows them to exploit good architectures as well as initializations. Note that, for CIFAR-10-C experiments, this was not the case; base learner losses did not have as sensitive a dependence on the initialization as they did on the architecture.

In Table 2, we compare the classification error and expected calibration error (ECE) of NES algorithms with the deep ensembles baseline for various ensemble sizes on Fashion-MNIST. Similar to the loss, NES algorithms also achieve smaller errors, while ECE remains approximately the same for all methods.

Table 2: Error and ECE of ensembles on Fashion-MNIST for different ensemble sizes $M$. Best values and all values within $95\%$ confidence interval are bold faced.

| | Classification Error (out of 1) | | | | | Expected Calibration Error (ECE) | | | | |
|---|---|---|---|---|---|---|---|---|---|---|
| $M$ | NES-RS | NES-RE | DeepEns (RS) | DeepEns (DARTS) | DeepEns (AmoebaNet) | NES-RS | NES-RE | DeepEns (RS) | DeepEns (DARTS) | DeepEns (AmoebaNet) |
| 3 | $0.074_{\pm 0.001}$ | $\mathbf{0.072}_{\pm 0.001}$ | $0.076_{\pm 0.001}$ | 0.077 | 0.077 | $0.007_{\pm 0.001}$ | $0.007_{\pm 0.002}$ | $0.008_{\pm 0.001}$ | **0.003** | 0.008 |
| 5 | $\mathbf{0.073}_{\pm 0.001}$ | $\mathbf{0.071}_{\pm 0.002}$ | $0.075_{\pm 0.001}$ | 0.077 | 0.074 | $\mathbf{0.005}_{\pm 0.001}$ | $\mathbf{0.005}_{\pm 0.001}$ | $\mathbf{0.006}_{\pm 0.001}$ | **0.005** | **0.005** |
| 10 | $0.073_{\pm 0.001}$ | $\mathbf{0.070}_{\pm 0.001}$ | $0.075_{\pm 0.001}$ | 0.076 | 0.073 | $\mathbf{0.004}_{\pm 0.001}$ | $0.005_{\pm 0.001}$ | $0.005_{\pm 0.001}$ | 0.006 | **0.005** |
| 30 | $0.073_{\pm 0.001}$ | $\mathbf{0.070}_{\pm 0.001}$ | $0.074_{\pm 0.001}$ | 0.075 | 0.073 | $\mathbf{0.004}_{\pm 0.001}$ | $\mathbf{0.004}_{\pm 0.002}$ | $\mathbf{0.004}_{\pm 0.001}$ | 0.008 | **0.004** |

### C.1.2 Entropy on out-of-distribution inputs

To assess how well models respond to completely out-of-distribution (OOD) inputs (inputs which do not belong to one of the classes the model can predict), we investigate the entropy of the predicted probability distribution over the classes when the input is OOD. Higher entropy of the predicted probabilities indicates more uncertainty in the model's output. For CIFAR-10 on the DARTS search space, we compare the entropy of the predictions made by NES ensembles with deep ensembles on two types of OOD inputs: images from the SVHN dataset and Gaussian noise. In Figure 10, we notice that NES ensembles indicate higher uncertainty when given inputs of Gaussian noise than deep ensembles but behave similarly to deep ensembles for inputs from SVHN.

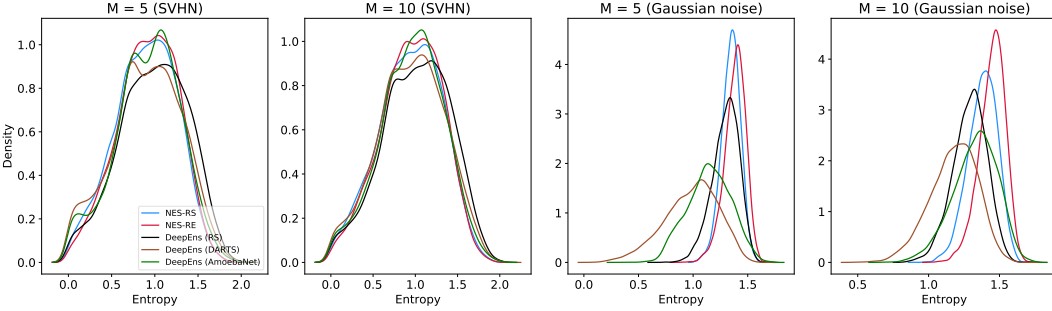

Figure 10: Entropy of predicted probabilities when trained on CIFAR-10 over the DARTS search space.

### C.1.3 Additional Results on CIFAR-10, CIFAR-100 and Tiny ImageNet

In this section, we provide additional experimental results on CIFAR-10, CIFAR-100 and Tiny ImageNet on the DARTS search space, complimenting the results in Section 5 as shown in Figures 12-18.

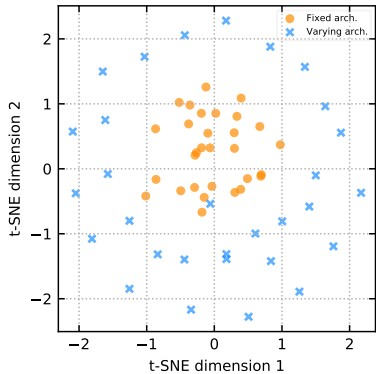

Figure 11: t-SNE visualization: predictions of base learners in two ensembles, one with fixed architecture and one with varying architectures.

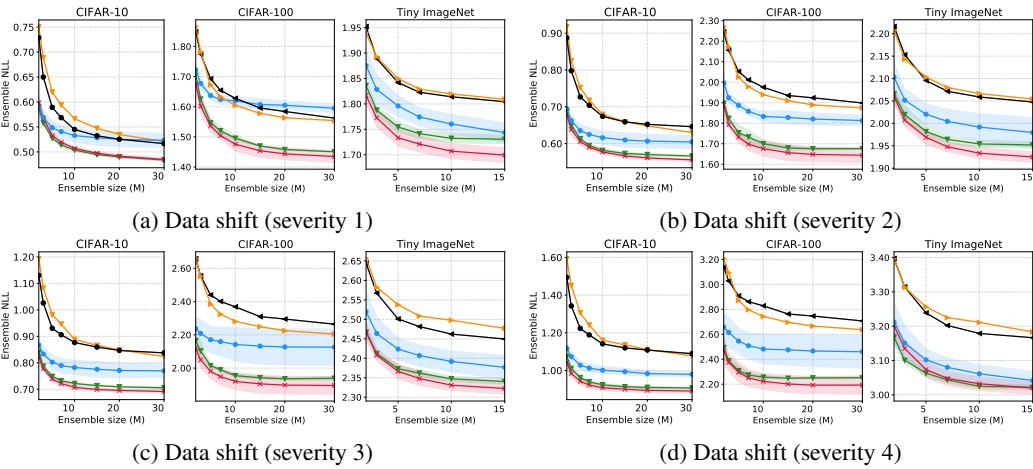

Figure 12: NLL vs. ensemble sizes on CIFAR-10, CIFAR-100 and Tiny ImageNet with varying dataset shifts (Hendrycks & Dietterich, 2019) over DARTS search space.

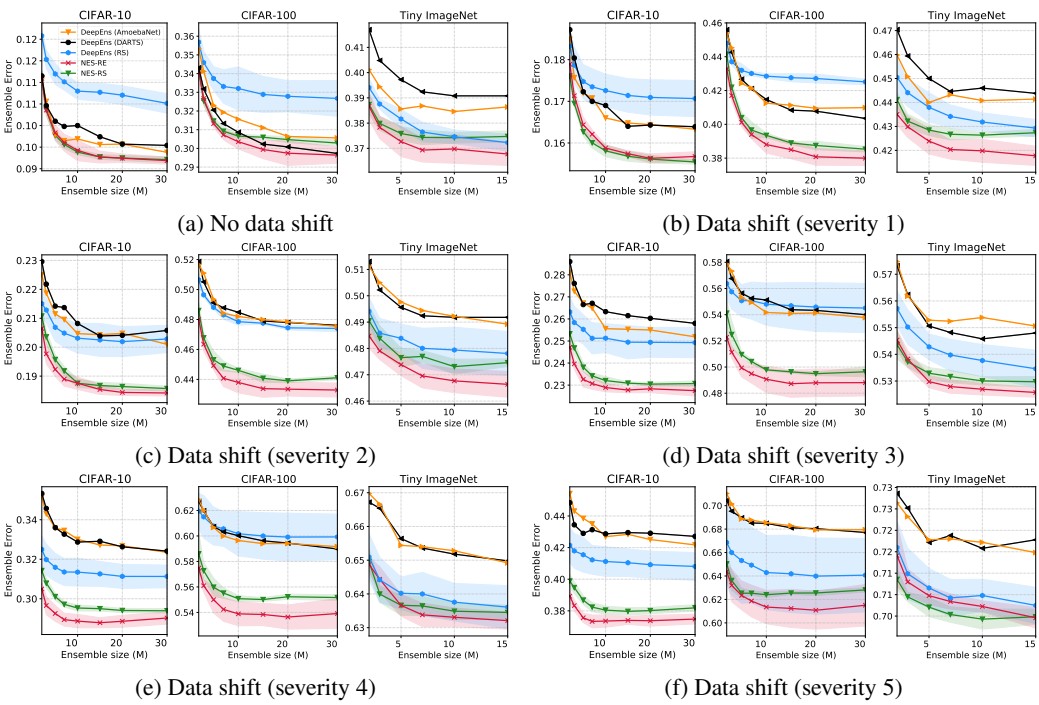

Figure 13: Classification error rate (between 0-1) vs. ensemble size on DARTS search space.

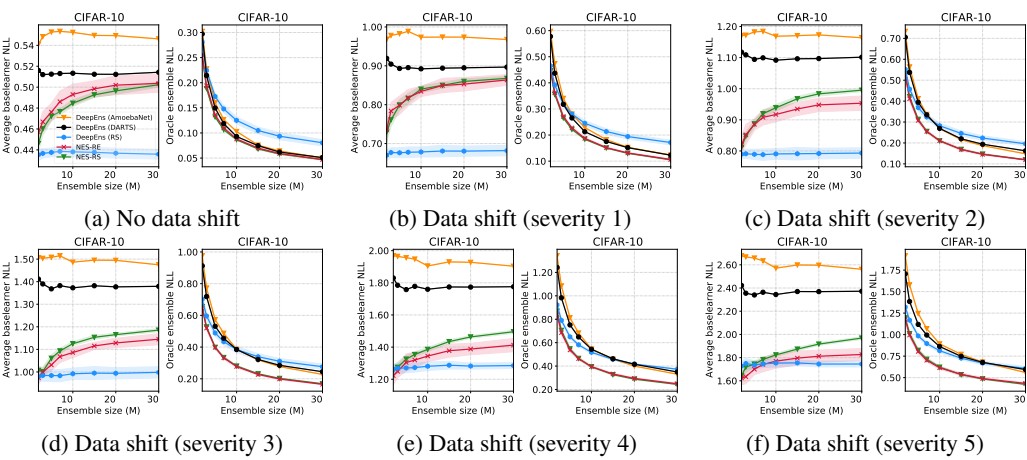

Figure 14: Average base learner and oracle ensemble NLL across ensemble sizes and shift severities on CIFAR-10 over DARTS search space.

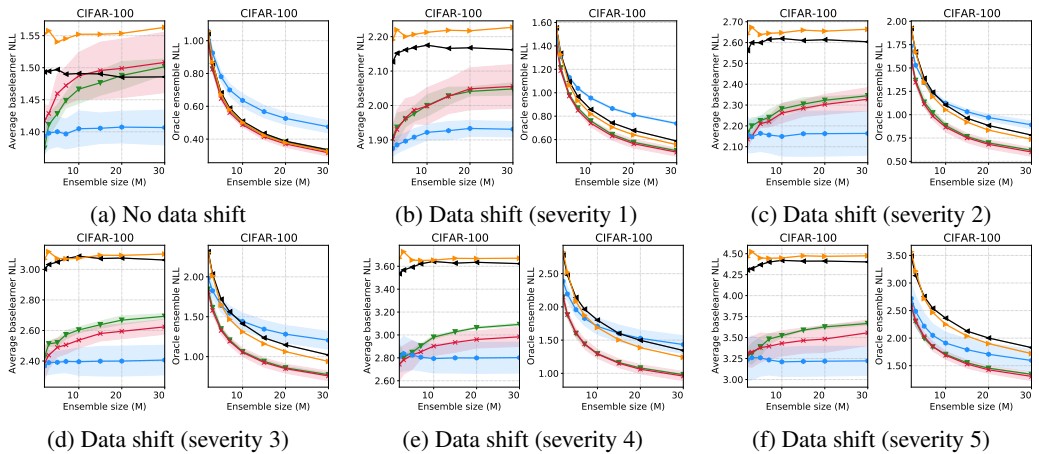

Figure 15: Average base learner and oracle ensemble NLL across ensemble sizes and shift severities on CIFAR-100 over DARTS search space.

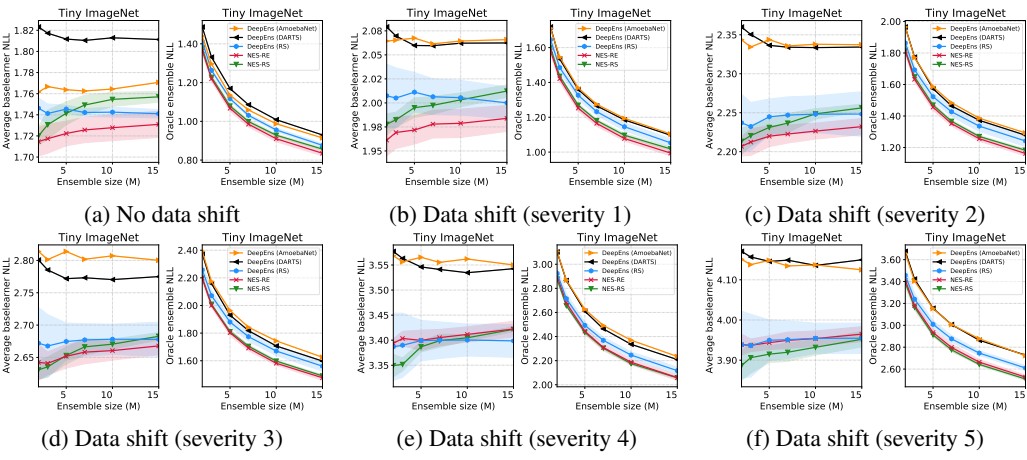

Figure 16: Average base learner and oracle ensemble NLL across ensemble sizes and shift severities on Tiny ImageNet over DARTS search space.

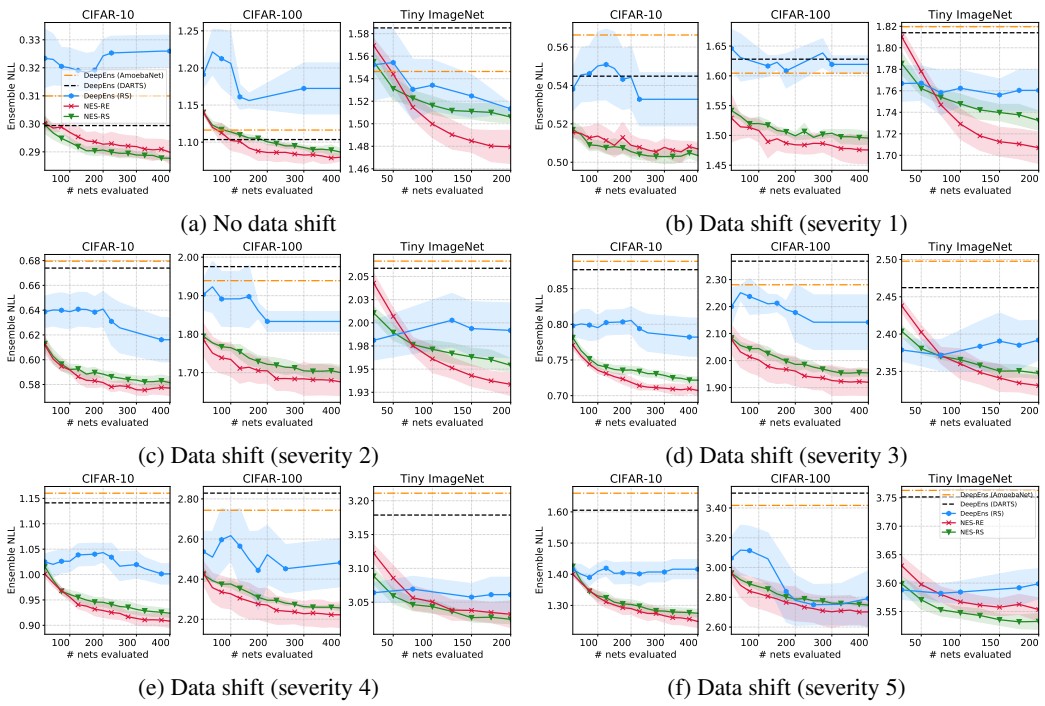

Figure 17: Ensemble NLL vs. budget $K$. Ensemble size fixed at $M = 10$.

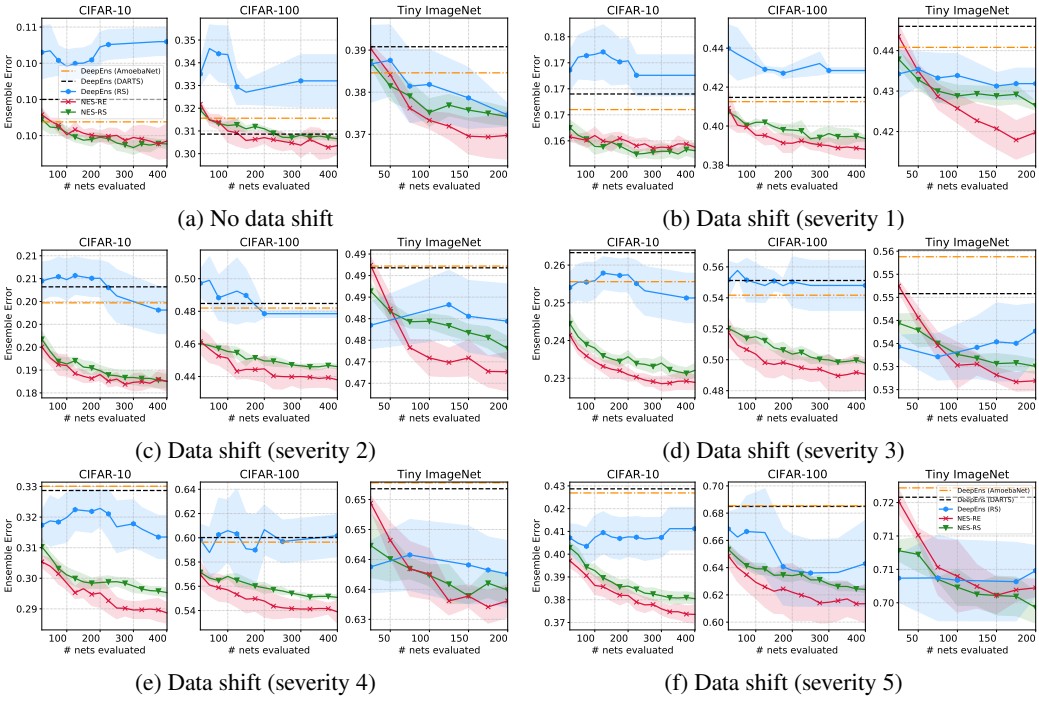

Figure 18: Ensemble error vs. budget $K$. Ensemble size fixed at $M = 10$.

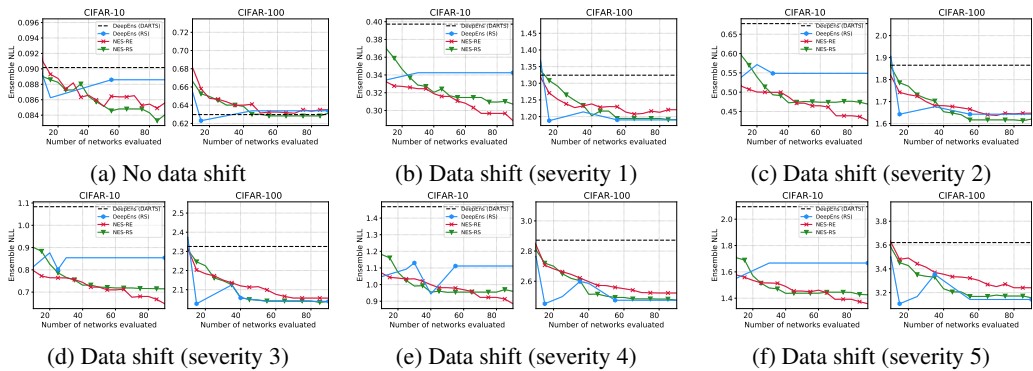

Figure 19: High fidelity NLL vs. budget $K$ on CIFAR-10 and CIFAR-100 with and without respective dataset shifts over the DARTS search space. Ensemble size is fixed at $M = 10$.

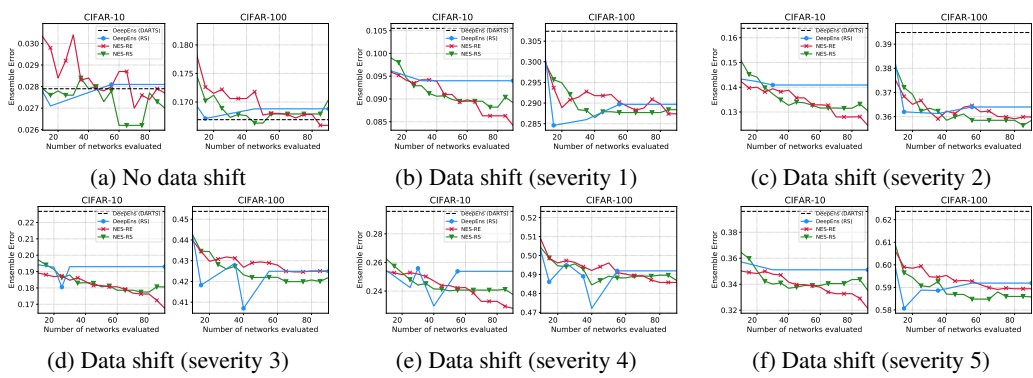

Figure 20: High fidelity classification error vs. budget $K$ on CIFAR-10 and CIFAR-100 with and without respective dataset shifts over the DARTS search space. Ensemble size is fixed at $M = 10$.

**Results on CIFAR for larger models.** In additional to the results on CIFAR-10 and CIFAR-100 using the settings described in Appendix B.3, we also train larger models (around 3M parameters) by scaling up the number of stacks cells and initial channels in the network. We run NES and other baselines similarly as done before and plot results in Figure 19 and 20 for NLL and classification test error with budget $K = 90$. As shown, NES algorithms tend to outperform or be competitive with the baselines. Note, more runs are needed including error bars for conclusive results in this case.

## C.2 ABLATION STUDY: NES-RE OPTIMIZING ONLY ON CLEAN DATA

We also include a variant of NES-RE, called NES-RE-0, in Figure 21. NES-RE and NES-RE-0 are the same, except that NES-RE-0 uses the validation set $\mathcal{D}_{\text{val}}$ without any shift during iterations of evolution, as in line 4 of Algorithm 2. Following the discussion in Appendix B.4, recall that this is unlike NES-RE, where we sample the validation set to be either $\mathcal{D}_{\text{val}}$ or $\mathcal{D}_{\text{val}}^{\text{shift}}$ at each iteration of evolution. Therefore, NES-RE-0 evolves the population without taking into account dataset shift, with $\mathcal{D}_{\text{val}}^{\text{shift}}$ only being used for the post-hoc ensemble selection step in line 9 of Algorithm 2.

As shown in the Figure 21, NES-RE-0 shows a minor improvement over NES-RE in terms of loss for ensemble size $M = 30$ in the absence of dataset shift. This is in line with expectations, because evolution in NES-RE-0 focuses on finding base learners which form strong ensembles for in-distribution data. On the other hand, when there is dataset shift, the performance of NES-RE-0 ensembles degrades, yielding higher loss and error than both NES-RS and NES-RE. Nonetheless, NES-RE-0 still manages to outperform the DeepEns baselines consistently. We draw two conclusions on the basis of these results: (1) NES-RE-0 can be a competitive option in the absence of dataset shift. (2) Sampling the validation set, as done in NES-RE, to be $\mathcal{D}_{\text{val}}$ or $\mathcal{D}_{\text{val}}^{\text{shift}}$ in line 4 of Algorithm 2 plays an important role is returning a final pool $\mathcal{P}$ of base learners from which `ForwardSelect` can select ensembles robust to dataset shift.

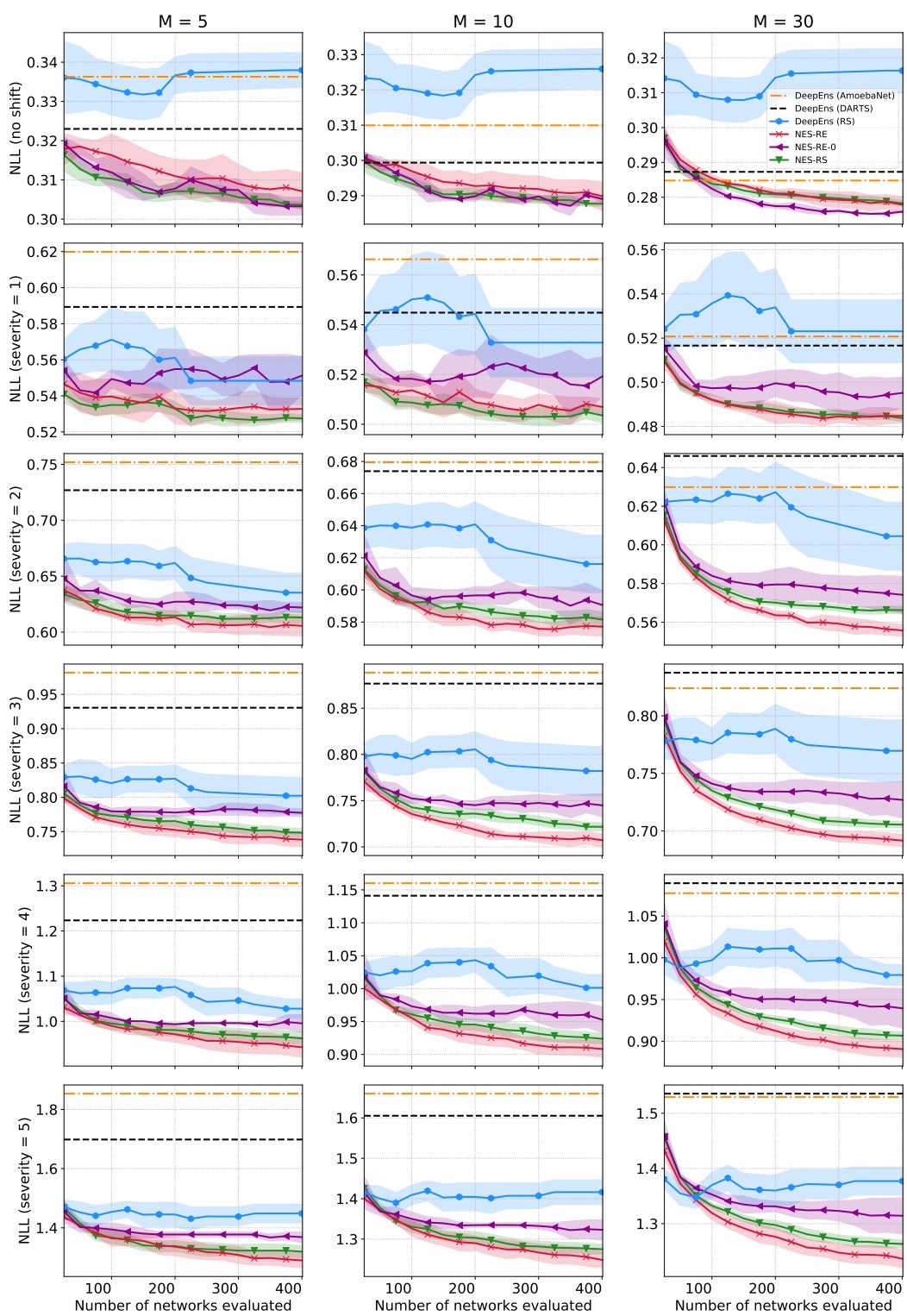

Figure 21: Results on CIFAR-10 (Hendrycks & Dietterich, 2019) with varying ensembles sizes $M$ and shift severity. Lines show the mean NLL achieved by the ensembles with 95% confidence intervals. See Appendix C.1.3 for the definition of NES-RE-0.

### C.3 WHAT IF DEEP ENSEMBLES USE ENSEMBLE SELECTION OVER INITIALIZATIONS?

Recall that NES algorithms differ from deep ensembles in two important ways: the ensembles use varying architectures and NES utilizes ensemble selection (i.e. `ForwardSelect` applied to $\mathcal{P}$) to pick the base learners. In this section, we conduct a study intended to investigate the following question: is the improvement offered by NES over deep ensembles only due to ensemble selection? In other words, we wish to isolate and understand the impact of varying architectures by comparing NES to a baseline that also incorporates ensemble selection into the construction of deep ensembles.

Using the DARTS search space on Tiny ImageNet, we empirically compare NES to the baselines "DeepEns + ES" which operate as follows. We optimize a fixed architecture for the base learners, train $K$ random initializations of it to form a pool and apply `ForwardSelect` to select an ensemble of size $M$ from the pool. This yields the three additional baselines DeepEns + ES (DARTS/AmoebaNet/RS) which correspond to optimizing the fixed architectures using the DARTS algorithm (DARTS), regularized evolution (AmoebaNet) and random search (RS).

The results indicate that NES outperforms or is at par with DeepEns + ES baselines, as shown in Table 3 and Figure 22. In particular, both NES algorithms outperform DeepEns + ES (DARTS/AmoebaNet). DeepEns + ES (RS) is the most competitive of the deep ensemble baselines, which is improved upon by NES-RE and is competitive with NES-RS. Also, as expected, deep ensembles *with* ensemble selection consistently perform better than their counterparts *without* ensemble selection.

Table 3 also includes the computational costs of each method measured in terms of the number of networks trained. For each DeepEns + ES baseline, we used a pool of size $K = 200$ (as with NES) from which the ensemble is selected. This cost comes in addition to the cost of optimizing the fixed base learner architecture prior to forming the pool. For instance, the architecture for DeepEns + ES (RS) is optimized by random search, selecting the best architecture by validation loss from a random sample of $K = 200$ (trained) architectures; this yields a total cost of 400 networks trained. This is twice the cost of NES algorithms which required training 200 architectures to form the pool.

### C.4 COMPARING NES TO ENSEMBLES WITH OTHER VARYING HYPERPARAMETERS

Since varying the architecture in an ensemble improves predictive performance and uncertainty estimation as demonstrated in Section 5, it is natural to ask what other hyperparameters should be varied in an ensemble. It is also unclear which hyperparameters might be more important than others. Note that concurrent work by Wenzel et al. (2020) has shown that varying hyperparameters such as $L_2$ regularization strength, dropout rate and label smoothing parameter also improves upon deep ensembles. While these questions lie outside the scope of our work and are left for future work, we conduct preliminary experiments to address them.

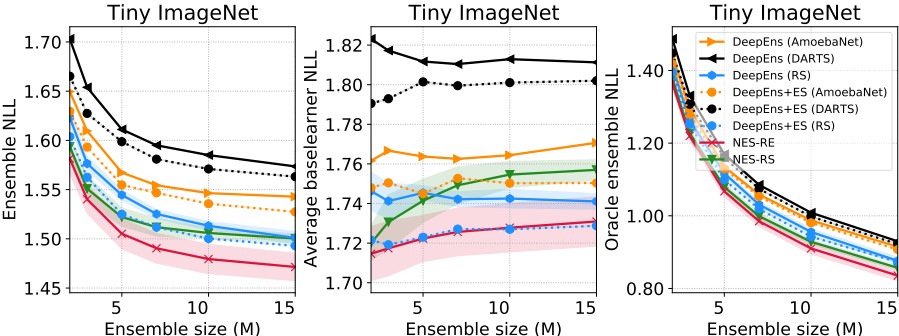

Figure 22: Loss vs. ensemble size for NES and deep ensembles (with/without ensemble selection over initializations). The left plot shows that NES-RE outperforms all other methods across ensemble sizes. The right plot shows that ensembles produced by NES algorithms also consistently have higher diversity (as indicated by smaller oracle ensemble loss). See Appendix C.3 for details.

Table 3: A comparison of NES to deep ensembles with ensemble selection over initializations for Tiny ImageNet over the DARTS search space with ensemble size $M = 10$. The computational costs are reports in terms of the number of networks trained (a typical network from this search space takes 3 hours to train on an NVIDIA RTX 2080Ti). The "arch" column indicates the number of architectures evaluated to find the architecture and the "ensemble" column indicates the number of architectures evaluated for building the ensemble. Note that for DARTS and AmoebaNet we convert the GPU hours for finding the architecture into number of networks trained by dividing by 3. See Appendix C.3 for details.

| Method | No dataset shift | | Dataset shift (severity 5) | | Cost (# nets trained) | |
|---|---|---|---|---|---|---|
| | NLL | Classif. Error (%) | NLL | Classif. Error (%) | Arch. | Ensemble |
| DeepEns (DARTS) | 1.59 | 39.08 | 3.75 | 71.58 | 32 | 10 |
| DeepEns + ES (DARTS) | 1.57 | 38.68 | 3.68 | 70.90 | 32 | 200 |
| DeepEns (AmoebaNet) | 1.55 | 38.46 | 3.76 | 71.72 | 25200 | 10 |
| DeepEns + ES (AmoebaNet) | 1.54 | 38.12 | 3.70 | 71.68 | 25200 | 200 |
| DeepEns (RS) | $1.51_{\pm 0.00}$ | $\mathbf{37.46}_{\pm 0.27}$ | $3.60_{\pm 0.03}$ | $70.48_{\pm 0.38}$ | 200 | 10 |
| DeepEns + ES (RS) | 1.50 | $\mathbf{36.98}$ | 3.55 | $\mathbf{70.10}$ | 200 | 200 |
| NES-RS | $1.51_{\pm 0.01}$ | $\mathbf{37.42}_{\pm 0.21}$ | $\mathbf{3.53}_{\pm 0.01}$ | $\mathbf{69.93}_{\pm 0.23}$ | | 200 |
| NES-RE | $\mathbf{1.48}_{\pm 0.01}$ | $\mathbf{36.98}_{\pm 0.57}$ | $3.55_{\pm 0.02}$ | $70.22_{\pm 0.13}$ | | 200 |

In this section, we consider two additional baselines working over the DARTS search space on CIFAR-10/100:

1. **HyperEns:** Optimize a fixed architecture, train $K$ random initializations of it *where the learning rate and $L_2$ regularization strength are also sampled randomly* and select the final ensemble of size $M$ from the pool using `ForwardSelect`. This is similar to `hyper ens` from Wenzel et al. (2020).

2. **NES-RS (depth, width):** As described in Appendix B.1, NES navigates a complex (non-Euclidean) search space of architectures by varying the cell, which involves changing both the DAG structure of the cell and the operations at each edge of the DAG. We consider a baseline in which we keep the cell fixed (the optimized DARTS cell) and only vary the width and depth of the overall architecture. More specifically, we vary the number of *initial channels* $\in \{12, 14, 16, 18, 20\}$ (width) and the number of *layers* $\in \{5, 8, 11\}$ (depth). We apply NES-RS over this substantially simpler search space of architectures as usual: train $K$ randomly sampled architectures (i.e. sampling only depth and width) to form a pool and select the ensemble from it.

The results shown in Figures 23 and Table 4 compare the two baselines above to DeepEns (DARTS), NES-RS and NES-RE.[4] As shown in Figure 23, NES-RE tends to outperform the baselines, though is at par with HyperEns on CIFAR-100 without dataset shift (Figure 23a). Under the presence of dataset shift (Figures 23b and 23c), both NES algorithms substantially outperform all baselines. Note that both HyperEns and NES-RS (depth, width) follow the same protocol as NES-RS and NES-RE: ensemble selection uses a shifted validation dataset when evaluating on a shifted test dataset. In terms of classification error, the observations are similar as shown in Table 4. Lastly, we view the diversity of the ensembles from the perspective of oracle ensemble loss in Figure 24. As in Section 5, results here also suggest that NES agorithms tend to find more diverse ensembles despite having higher average base learner loss.

---

[4]Note that runs of DeepEns (DARTS), NES-RE and NES-RS differ slightly in this section relative to Section 5, as we tune the learning rate and $L_2$ regularization strength for each dataset instead of using the defaults used in Liu et al. (2019). This yields a fair comparison: HyperEns varies the learning rate and $L_2$ regularization while using a fixed, optimized architecture (DARTS), whereas NES varies the architecture while using fixed, optimized learning rate and $L_2$ regularization strength.

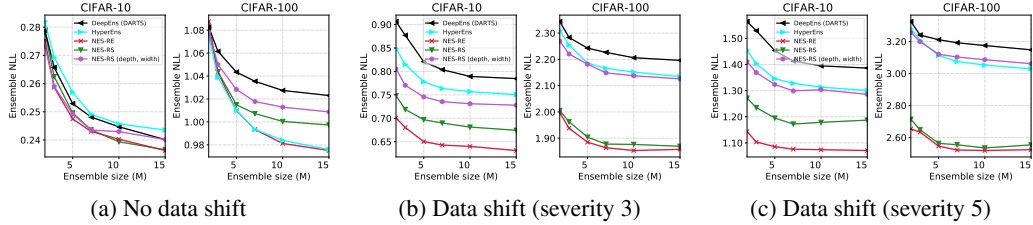

(a) No data shift     (b) Data shift (severity 3)     (c) Data shift (severity 5)

Figure 23: Plots show NLL vs. ensemble sizes comparing NES to the baselines introduced in Appendix C.4 on CIFAR-10 and CIFAR-100 with and without respective dataset shifts (Hendrycks & Dietterich, 2019).

Table 4: Classification errors comparing NES to the baselines introduced in Appendix C.4 for different shift severities and $M = 10$. Best values are bold faced.

| Dataset | Shift Severity | DARTS search space | | | | |
|---------|----------------|---------------------|----------|----------------------------|--------|--------|
| | | DeepEns (DARTS) | HyperEns | NES-RS (depth, width) | NES-RS | NES-RE |
| C10 | 0 | 8.2 | 8.1 | 8.0 | 8.0 | **7.7** |
| | 3 | 25.9 | 25.0 | 24.1 | 22.5 | **21.5** |
| | 5 | 43.3 | 40.8 | 42.1 | 38.1 | **34.9** |
| C100 | 0 | 28.8 | **28.1** | 28.8 | 28.4 | 28.4 |
| | 3 | 54.0 | 52.7 | 53.1 | 48.9 | **48.5** |
| | 5 | 68.4 | 67.2 | 67.9 | 61.3 | **60.7** |

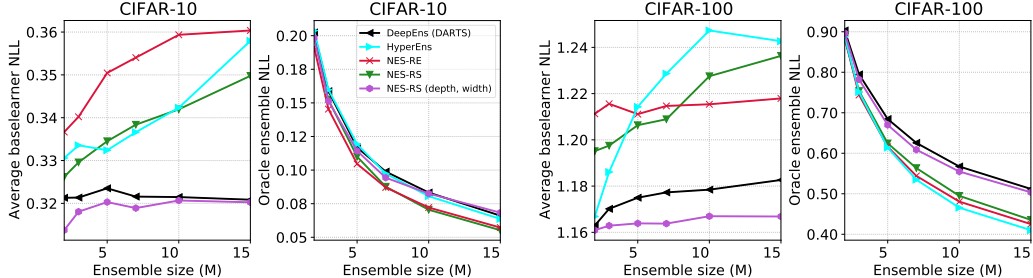

Figure 24: Average base learner loss and oracle ensemble loss for NES and the baselines introduced in Appendix C.4 on CIFAR-10 and CIFAR-100. Recall that small oracle ensemble loss generally corresponds to higher diversity.

