# OpenReview forum: "Neural Ensemble Search for Uncertainty Estimation and Dataset Shift"
_ICLR.cc/2021/Conference — Reject_

### Official Review · AnonReviewer4 · 2020-10-27
**A promising solution in an interesting topic. Some modifications are needed to be more convincing.**

**Rating:** 6
**Confidence:** 4

**Review:**

The authors addressed my concerns in the rebuttal. I have raised my score.

Summary:

This paper combines AutoML techniques and deep ensembles to improve the ensemble diversity so that it improves the entire ensemble quality in both in- and out-of-distribution dataset. The authors made a study on two possible AutoML methods which can be combined with ensembles: 1. Random search & 2. Regularized mutation. The empirical results showed that the proposed NES methods outperform commonly selected baselines.

Pros:

How to improve ensemble diversity is one of the core topics on the way to better ensemble performance (in terms of both accuracy and uncertainty metrics). Many previous research works focused on 1. Build efficient ensembles while remaining reasonable diversity; 2. Improve ensemble diversity by exploring the hyper-parameter space. In practice, it is standard to ensemble neural networks with different depths or widths to improve diversity and hence the ensemble performance. However, as far as I know, there is no guidance or automatic mechanism in ensembling neural networks with different architectures. Thus, the problem this paper aims to tackle is significant and it will benefit the research community.

The authors did a self-contained introduction on ensembles & uncertainty and AutoML. The coverage on mutation AutoML is limited but this is still reasonable due to the page constraint. The motivation of why we want to combine AutoML and deep ensembles is clearly stated in section 3.2. Figure 1 demonstrates the effectiveness of various architectures in promoting ensemble diversity.

The empirical evaluation ranges across CIFAR dataset and ImageNet. It also includes calibration performance under dataset shift (uncertainty estimation on out-of-distribution dataset), which is the common benchmark to evaluate an ensemble's performance. The authors made comparisons to several reasonable baselines. The improvement of the proposed NES-RE/RS method over fixed architecture ensembles is consistent and significant.


Cons:

Figure 1 demonstrates the motivation behind this work. To be more convincing, it can be supplemented with the predictive disagreement on the testset or the averaged KL divergence between the predictive distribution among ensemble members. Moreover,  the figure compares diversity between ensembles with different architectures and ensembles with random seeds. For a more comprehensive study, the figure can include a study on ensembles with different hyper-parameters. A more interesting baseline I will mention below is an ensemble with different depths.

The baselines considered in this paper only include ensembles with a fixed architecture. It would be more convincing if the authors can include other baselines which include ensembles with different architectures (without neural architecture search). For example, one naive baseline would be ensembling DeepEns(Optimal) with different depths (fully trained independently). This highlights the need for neural architecture search. It also helps to understand how much diversity in architecture (among ensemble members) we need in order to achieve desired diversity in ensemble predictions. Additionally, it is encouraged to compare to hyper-parameter ensembles. This uncovers the question of which axis (hyper-parameter & architectures) is more effective in promoting an ensemble’s performance.

Another missing part in this paper is the cost analysis of NES and how much does it increase compared to deep ensembles. Both NES-RS and NES-RE require training after sampling one neural architecture. This leads to a non-trivial computational overhead compared to traditional deep ensembles.

In section 4.3, it mentioned that a proportion of validation data encapsulates the belief about test-time shift. The authors didn’t mention whether the same protocol is applied to baselines (DeepEns). This leads to a question that is the improvement on out-of-distribution calibration coming from NES or shifted validation set. I consider this is a minor issue because Figure 21 in the appendix also shows that the clear improvement without shifted validation data.

Table 1 shows that larger ensemble size leads to worse ensembling performance. This is against our general intuition in deep ensembles where more ensemble members lead to better performance. My guess is more ensemble members leads to optimization difficulties in NES. I expect to see more discussion on this observation.

Overall, the authors propose a compelling solution to automatically design the neural network architectures in deep ensembels. However,  the cons slightly outweight the pros in this version.

---

> ### Author Response · Authors · 2020-11-18
> **Response to AnonReviewer4 (Part 1/2)**
>
> Many thanks for your detailed, helpful review and for appreciating our work. We address your concerns below by incorporating your suggestions into our work and hope that you will consider updating your score:
>
> 1. **“To be more convincing, [Figure 1] can be supplemented with the predictive disagreement on the test set”**: Thanks for this suggestion! We have added a comparison of the predictive disagreement in an ensemble with fixed architecture vs an ensemble with varying architectures in the last paragraph of Section 3.2. The results (11.88% vs. 10.51% disagreement for varying vs. fixed architecture ensembles respectively) show that varying the architecture also yields higher predictive disagreement, i.e. higher diversity.
>
> 2. **“The baselines considered in this paper only include ensembles with a fixed architecture. It would be more convincing if the authors can include other baselines which include ensembles with different architectures (without neural architecture search). For example, one naive baseline would be ensembling DeepEns(Optimal) with different depths (fully trained independently). This highlights the need for neural architecture search. It also helps to understand how much diversity in architecture (among ensemble members) we need in order to achieve desired diversity in ensemble predictions. Additionally, it is encouraged to compare to hyper-parameter ensembles. This uncovers the question of which axis (hyper-parameter & architectures) is more effective in promoting an ensemble’s performance.”**: Thank you for this suggestion! We agree that both baselines you suggested are reasonable to compare, therefore we have added Appendix C.4 comparing NES to them. Note that hyperparameter ensembles [1] is concurrent work to ours (ours posted on arXiv a week earlier), and while both papers show that varying particular hyperparameters (note architecture is a hyperparameter) is beneficial, the question of which hyperparameters one should vary remains open and is left for future work. Nonetheless, we perform a comparison to two new baselines in Appendix C.4: 1. HyperEns: ensembles with a fixed, optimized architecture but varying learning rates and L2 regularization strengths, and 2. NES-RS (depth, width): ensembles with architectures varying only in terms of width and depth, keeping the cell fixed. The results show that NES tends to improve upon these baselines. Please refer to Appendix C.4 for details.
>
> 3. **“Another missing part in this paper is the cost analysis of NES”**: Thanks for this important comment! We have added a paragraph in Section 5 comparing the computational cost of NES vs. baselines, explaining why NES is not necessarily more costly than DeepEns. In summary, DeepEns baselines have two costs (which are subsumed into one cost in NES as explained below): finding an optimized, fixed base learner architecture and training M initializations of it. While both steps incur a cost, the former step can be extremely costly (e.g. where it is not clear what base learner architecture is best given a set of choices, requiring the use of random search or a typical NAS algorithm to find an optimized architecture). A concrete example from our paper is the deep ensemble made using AmoebaNet, which is an architecture found using a regularized evolution run that required 3150 GPU days. As you mention, “as far as I know, there is no guidance or automatic mechanism in ensembling neural networks with different architectures”; NES combines the architecture search and ensembling components, with its only cost being the training of K architectures to form the pool. Also, see Table 3 regarding costs in Appendix C.3.

---

> > ### Author Response · Authors · 2020-11-18
> > **Response to AnonReviewer4 (Part 2/2)**
> >
> > 4. **“The authors didn’t mention whether the [shifted validation set] protocol is applied to baselines (DeepEns). This leads to a question that is the improvement on out-of-distribution calibration coming from NES or shifted validation set. I consider this is a minor issue because Figure 21 in the appendix also shows that the clear improvement without shifted validation data.”**: Our apologies that this was unclear. We now explain the use of the shifted validation data for the baselines.
> > First, for evaluations on test data with shift (e.g. Figures 3b and 4b), for DeepEns (RS) there are many architectures we can choose a fixed one from, and we indeed choose this to minimize loss on the shifted validation set from a random sample, i.e. random search (see paragraph on NLL in Section 5). However, for DeepEns (DARTS/AmoebaNet), this is different, since there we just use the fixed architecture found in the DARTS paper / in the AmoebaNet paper. So, in this case, the same architecture is used when evaluating on test data with/without shift. Second, as you correctly point out, when evaluating on test data without shift (e.g. Figures 3a and 4a), none of the methods utilize a shifted validation set and we see a “clear improvement” of NES over baselines. Third, in response to AnonReviewer5, during the author response phase, we added a new baseline DeepEns+ES (DeepEns + Ensemble Selection) in which all deep ensembles utilize shifted validation data (see Appendix C.3). In this case, we exactly follow the protocol used for NES: ensemble selection uses shifted validation data when evaluating on shifted test data. In particular, also the DARTS/AmoebaNet deep ensembles in this baseline now *do* use the shifted validation data. We nevertheless find NES continues to outperform this baseline.
> >
> > 5. **“Table 1 shows that larger ensemble size leads to worse ensembling performance. This is against our general intuition in deep ensembles where more ensemble members lead to better performance. My guess is more ensemble members leads to optimization difficulties in NES. I expect to see more discussion on this observation.”**: This is due to a misunderstanding. Please note that the two sub-tables in Table 1 show results on different search spaces using different model sizes and training routines (sorry, we have fixed the captions to make this clear!), so they are incomparable. As you point out, larger ensembles do indeed lead to better performance, and this is consistently true in our detailed experiments with growing ensemble size as shown in Figure 3.
> >
> > We hope that the clarification of the simple misunderstanding in 5., comparison with respect to predictive disagreement and the addition of the additional baselines might convince you to increase your score. We are looking forward to any additional questions that may have remained unanswered.
> >
> > -- References --
> > [1] Wenzel et al. Hyperparameter Ensembles for Robustness and Uncertainty Quantification. In NeurIPS 2020

---

### Official Review · AnonReviewer1 · 2020-10-27
**Important problem but the solution lacks novelty**

**Rating:** 4
**Confidence:** 5

**Review:**

The paper proposes creating diverse ensembles of neural networks using an evolutionary method for finding base learners with high performance as well as mutual diversity. The selected base learners are then aggregated for an ensemble using a known method for ensemble selection. The paper is generally well written and addresses a relevant problem of constructing ensembles while training neural networks instead of building models first, independently, and later constructing ensembles.

Having said that, the paper lacks a significant contribution. The second phase (Ensemble Selection) of the proposed method is essentially the algorithm from Rich Caruana et al. 2004. The first phase of Pool Building suggests either a random generation or alternatively a vaguely described evolutionary method, lacking details or analysis.  It is not clear how exactly is the random initialization of architectures performed. Do you randomly select from a set of seed architectures or randomly create (i.e., random number of layers, random number of units, random activation functions, random initial weights, etc.)?

Growing from a random neural architectures to multiple highly performing (besides being mutually diverse) through single permutations upon model training and evaluation seems like an expensive process. An evolutionary approach in such a manner seems in efficient. Can you report the time taken for some of the reported cases in the evaluation? The ensemble methodology of unweighted averaging is fairly naive. What was the reason to select this one particularly?

Contribution #1 (page 2) isn't really a contribution. It is common knowledge amongst practitioners. The proposed method can lead to overfitting because the search seems to be based on a fixed set for evaluation.

Regarding evaluation -- Can you explain how that is addressed? Have you evaluated your method on a broader variety of datasets? Can you confirm that the test data used for search/optimization is different than the one used for measuring reported performance? Did you consider comparing it other methods such as Tao 2019 (mentioned below) ?

This paper can improve its literature survey by citing more directly relevant work in ensemble search using diversification. Here are few examples of more sophisticated ensemble evolution work, not necessarily for a DL base learner, but relevant nonetheless:
-Bhowan, et al. 2013. Evolving diverse ensembles using genetic programming for classification with unbalanced data. Trans. Evol. Comp
-Khurana et al. 2018. Ensembles with Automated Feature Engineering. AutoML at ICML.
-Olson et al. 2019. TPOT: A Tree-Based Pipeline Optimization Tool for Automating Machine Learning. Automated ML.
-Tao, 2019. Deep Neural Network Ensembles. Machine Learning, Optimization, and Data Science.
-Yao et al., 2008. Evolving artificial neural network ensembles, in IEEE Computational Intelligence Magazine

Overall, it is a good problem, but this paper falls well short of the threshold.
Update: I thank the authors for their response. Some justifications are provided and for that I will change my score. Overall, the paper still needs work.

---

> ### Author Response · Authors · 2020-11-18
> **Response to AnonReviewer1**
>
> Thank you for your comments and the references. We have seen similar remarks previously and a number of them are addressed in our work; we refer to the appropriate sections.
>
> 1. **“ensemble selection... is essentially Caruana et al. (2004)”**: We cited Caruana et al. (2004) in explaining our choice of forward selection on pg. 4 and in the related work section, and we do not claim forward selection to be a contribution.
>
> 2. **“vaguely described evolutionary method, lacking details or analysis”**: Can you please specify what you felt is lacking in our description of NES-RE? We have described NES-RE in Section 4.2 and Figure 2. We have also provided pseudocode in Algorithm 2, implementation and scalability details in Appendix B.4 and code in the supplementary material.
>
> 3. **“Do you randomly select from a set of seed architectures or randomly create”**: The architecture search spaces and how we randomly sample architectures are described in Appendices B.1 and B.3. In short, architectures in the DARTS search space are specified by cells which are DAGs where each edge represents an operation (e.g. max pooling, separable convolution). We sample both the structure of the cell and the operations at each edge.
>
> 4. **“ensemble methodology of unweighted averaging is fairly naive”**: Despite it being simple and “naïve”, unweighted averaging is used in deep ensembles (and works well), and in order to isolate and evaluate the impact of varying architectures, we also used unweighted averaging. Note that multiple popular neural network ensembling techniques also use unweighted averaging (e.g. deep ensembles, snapshot ensembles, fast geometric ensembling etc.) Nonetheless, more sophisticated ensemble combination methods could readily be used with NES.
>
> 5. **“The proposed method can lead to overfitting”**: While it is difficult to guarantee no overfitting, in our experiments, we did not experience evidence for overfitting during ensemble selection despite using a fixed validation set. This is evident in Figures 4, 7, 17, since test performance of NES improves with increasing pool size/budget.
>
> 6. **“Regarding evaluation -- Can you explain how that is addressed? Have you evaluated your method on a broader variety of datasets? Can you confirm that the test data used for search/optimization is different than the one used for measuring reported performance?”**: Section 5 evaluates NES on 5 datasets (FMNIST, CIFAR-10, CIFAR-100, ImageNet-16-120 and Tiny ImageNet) and 2 architecture search spaces (DARTS search space and NAS-Bench-201), using 3 metrics (NLL, classification error, ECE) including when there is test-time dataset shift. Regarding the use of test data, yes, “unless stated otherwise, all evaluations are on the test dataset” (section 5), and Algorithms 1-2 only use D_train, D_val. Using suggestions from reviewers, we have also added comparisons to new baselines in Appendices C.3 and C.4. Overall, we believe that NES’ empirical performance has been extensively evaluated, as also noted by AnonReviewer4: *“The authors made comparisons to several reasonable baselines. The improvement of the proposed NES-RE/RS method over fixed architecture ensembles is consistent and significant.”*
>
> 7. **“Did you consider comparing it to other methods such as Tao 2019?”**: Thank you for the reference. We have not compared to Tao 2019, because this seems akin to a boosting-based approach which requires training base learners sequentially (at least, partially).  First, this is time-consuming for large networks and ensemble sizes (e.g.  size 30 in our experiments) in contrast with randomization-based approaches for ensembling, such as NES and deep ensembles, which are readily parallelized. We have also been unable to find code implementation for Tao 2019. Second, in the context of predictive uncertainty and dataset shift, ensemble diversity due to randomization-based approaches reduces overconfident predictions by individual baselearners. It is unclear this benefit would be retained by boosting-based approaches which optimize solely for predictive performance on in-distribution data.

---

### Official Review · AnonReviewer2 · 2020-10-30
**Interesting idea but more experimentation needed**

**Rating:** 4
**Confidence:** 5

**Review:**

The paper explores whether one can use Architecture Search to enhance ensemble diversity. They start with the observation that embeddings generated by different architectures (for multiple different initialization per architecture) are well separated from each other. They then try out a couple of architecture search methods to find ensembles with diverse architectures that minimize the loss.

While the novelty is incremental, I like the idea in general. My main objection is that critical baselines are not compared with. Ensemble diversity is a well explored topic with multiple easy to implement regularizations to increase diversity [1, 2, 3] and several more that should be compared with.

[1] “Maximizing Overall Diversity for Improved Uncertainty Estimates in Deep Ensembles” by S Jain, G Liu, DK Gifford (AAAI 2020)
[2] “Ensemble learning via negative correlation” by Y. Liu, X. Yao (Neural Networks 1999)
[3] “Uncertainty in Neural Networks: Approximately Bayesian Ensembling” by Pearce et al (AISTATS 2020)

---

> ### Author Response · Authors · 2020-11-18
> **Response to AnonReviewer2**
>
> Many thanks for taking the time to read our paper and for your feedback. Below we address the reviewer’s main concerns:
>
> 1. **"While the novelty is incremental, I like the idea in general."**: Thanks for your kind words! Regarding novelty: to our knowledge, ensembles with varying architectures have not previously been considered in the context of uncertainty calibration and dataset shift, and our work is the first to utilize ideas from NAS to build algorithms for automatically selecting the architectures and to show that this yields improvements in an empirical evaluation using state-of-the-art NAS search spaces.
>
> 2. **"My main objection is that critical baselines are not compared with. Ensemble diversity is a well explored topic with multiple easy to implement regularizations to increase diversity [1, 2, 3] and several more that should be compared with."**: Thank you for pointing us to these papers (added to our related work!). We do believe that we use meaningful baselines, which was also stated by AnonReviewer4: *"The authors made comparisons to several reasonable baselines."* A central aim of our work is to investigate the impact of varying architectures in an ensemble from the perspective of uncertainty estimation and dataset shift; we chose our baselines in accordance with this aim. Deep ensembles which keep the architecture fixed are therefore a natural baseline where the architecture is optimized and chosen from the same search space over which NES operates (e.g. DARTS and AmoebaNet architectures). Crucially, the training pipeline is identical for base learners in NES and deep ensembles. Diversity regularizing methods such as [1], [2] and [3] can be implemented on top of NES ensembles (i.e. train the selected architectures with diversity regularizing penalties). Also, note that work such as [4] have compared deep ensembles to multiple ensembling techniques (including snapshot ensembles, fast geometric ensembling, SWA-Gaussian, cyclical SGLD and dropout), showing that *“most of the popular ensembling techniques require averaging predictions across dozens of samples (members of an ensemble), yet are essentially equivalent to an ensemble of only few independently trained models.”* Based on the results of that paper, we believe deep ensembles form a difficult-to-beat baseline.
>
> -- References --
> [1] “Maximizing Overall Diversity for Improved Uncertainty Estimates in Deep Ensembles” by S Jain, G Liu, DK Gifford (AAAI 2020)
> [2] “Ensemble learning via negative correlation” by Y. Liu, X. Yao (Neural Networks 1999)
> [3] “Uncertainty in Neural Networks: Approximately Bayesian Ensembling” by Pearce et al (AISTATS 2020)
> [4] “Pitfalls of in-domain uncertainty estimation and ensembling in deep learning” by Ashukha et al. (ICLR 2020)

---

> > ### Comment · AnonReviewer2 · 2020-11-23
> > **Re:Author Response**
> >
> > Thank you for the response. I agree that diversity methods can be implemented on top of NES but I don't think it's clear whether NES+diversity methods would give more over just diversity methods so I think either measuring NES+diversity methods or a direct comparison of NES and diversity methods is important.

---

> > > ### Author Response · Authors · 2020-11-25
> > > **Reply regarding baselines**
> > >
> > > Thank you for your reply! While we agree that adding explicit diversity regularizers on top of NES is an interesting avenue to explore, we emphasize that this lies outside the scope of the questions we aimed to explore in our work. To answer our original question of the impact of varying architectures in deep ensembles, we compared to various variants of deep ensembles and ensembles with other varying hyperparameters (deep ensembles with different SOTA base learner architectures, with/without ensemble selection over initializations, ensembles with varying non-architectural hyperparameters, ensembles varying only in terms of depth/width). In discussion with other reviewers, we agreed that some further baselines were important to compare to, and we added those to Appendices C.3 and C.4. There are many baselines one can compare to, but we chose ones that provide the most insight into our original question. NES does not alter the loss function or include explicit diversity regularization such as NCL [1] and MOD [2] do. Also note that [3] only compare to deep ensembles and a variant of their method of anchored ensembles (i.e. “regularized” NNs which anchor all base learners to 0).
> > >
> > > -- References --
> > >
> > > [1] “Ensemble learning via negative correlation” by Y. Liu, X. Yao (Neural Networks 1999)
> > > [2] “Maximizing Overall Diversity for Improved Uncertainty Estimates in Deep Ensembles” by S Jain, G Liu, DK Gifford (AAAI 2020)
> > > [3] “Uncertainty in Neural Networks: Approximately Bayesian Ensembling” by Pearce et al (AISTATS 2020) [4] “Pitfalls of in-domain uncertainty estimation and ensembling in deep learning” by Ashukha et al. (ICLR 2020)

---

### Official Review · AnonReviewer5 · 2020-11-05
**Simple and interesting method but the important experiment is missing**

**Rating:** 5
**Confidence:** 4

**Review:**

The paper suggests a new approach to the construction of ensembles of deep neural networks (DNN). Unlike previous methods which usually deal with multiple DNNs of same structure authors propose to form an ensemble of networks with different architecture. The main claim is that using diverse architectures increases diversity and hence the quality of predictions. To find the best architectures they use methodology inspired by neural architecture search (NAS) in particular random search and regularized evolution. The method for neural ensemble search (NES) is algorithmically simple although computationally hard. On several experiments the authors show NES outperforms standard deep ensembles formed from networks with same (even optimal) structure both in terms of test NLL and in terms of uncertainty estimation under domain shift.

Pros.
Nice idea
Simple algorithm


Cons.
My main point for the criticism is the lack of experiment which I find to be crucially important namely the comparison aganist deep ensemble of DNNs with same architecture to which ForwardSelect procedure has been applied. Train P DNNs with same architecture then perform ForwardSelect routine to take the best K of them and compare your method with such deep ensemble. Currently the authors only compare their method with deep ensembles to which no special selection procedure was applied. This causes bias and it is not clear whether the improvement in NES is due to the usage of different architectures or due to the selection procedure which encourages diversity in resulting ensemble.

P.S. Please correct me if I misunderstood the last point. I have read the corresponding part twice and found no evidence that you're using ForwardSelection when analysing the performance of ensembles of DNNs with same architecture.

====UPDATE===
My concerns were partly addressed in author's response so I have raised my score to 5.

---

> ### Author Response · Authors · 2020-11-18
> **Response to AnonReviewer5**
>
> Thank you for your feedback. Below we address the reviewer’s main concern.
>
> 1. **"My main point for the criticism is the lack of experiment which I find to be crucially important namely the comparison against deep ensemble of DNNs with same architecture to which ForwardSelect procedure has been applied. Train P DNNs with same architecture then perform ForwardSelect routine to take the best K of them and compare your method with such deep ensemble. Currently the authors only compare their method with deep ensembles to which no special selection procedure was applied. This causes bias and it is not clear whether the improvement in NES is due to the usage of different architectures or due to the selection procedure which encourages diversity in resulting ensemble."**: Thank you for this suggestion! We agree this is an important study to gain insight into the improvement from NES, and we have added this ablation to our work in Appendix C.3. As you described, we compare to additional deep ensemble baselines (called “DeepEns + ES”) which select the ensemble from a pool of trained random initializations of a fixed, optimized architecture. Our results show that NES algorithms continue to outperform these baselines. Also, note that the cost of “DeepEns + ES” baselines is substantially higher at the ensembling stage than usual deep ensembles, as we now train a pool of random initializations instead of just M random initializations (M = ensemble size). In fact, the total cost ends up becoming larger than NES, since for DeepEns + ES we first need to find a good architecture and then train it multiple times to form a pool (as in NES). Appendix C.3 contains further discussion on these points.
>
> We hope the reviewer will consider increasing their score, as we have added the experiment they suggested. We welcome any questions.

---

> > ### Comment · AnonReviewer5 · 2020-11-23
> > **Thanks for the experiment**
> >
> > Thanks for the experiment! it shows that architecture search really matters although it seems that forwardselection procedure does the most of job even when the architecture is fixed. I am raising my score to 5.

---

> > > ### Author Response · Authors · 2020-11-24
> > > **Further comments on the experiment in Appendix C.3**
> > >
> > > Thank you! We are glad you agree that the experiment in Appendix C.3 indicates varying the architectures in crucial for the improvements in NES.
> > >
> > > **“forwardselection procedure does the most of job even when the architecture is fixed”**:
> > > Actually, Appendix C.3 suggests the opposite: the majority of gain in performance is due to ensembling and not ensemble selection (i.e. ForwardSelect) for the baselines “DeepEns + ES”. More specifically, Figure 22 shows that single model NLL for the DARTS architecture is on average 1.8-1.82. A deep ensemble (i.e. without ensemble selection) of (say) size 10, brings this down to around 1.59, and adding ensemble selection on top of that reduces the loss to around 1.57. While it is beneficial to perform ensemble selection, the additional gain it offers over ensembling is relatively small. (The same applies to DeepEns + ES (RS/AmoebaNet).) Separately, also note that NES-RE has a lower ensemble NLL even though DeepEns + ES (RS) and NES-RE have very similar average base learner performance, which reaffirms the importance of varying architectures. Does this clarify your concern?

---

### Author Response · Authors · 2020-11-18
**Reply to all reviewers about changes**

Thank you to all reviewers for their feedback and important suggestions. We have posted individual responses. We highlight the main changes made to our paper here:

1. We have added a new baseline in Appendix C.3 comparing NES to deep ensembles with ForwardSelect applied over a pool of random initializations of the fixed architecture, finding that NES also outperforms the resulting ensembles. (Suggested by *AnonReviewer5*)

2. We have added Appendix C.4 where we compare NES to ensembles with other hyperparameters being varied, also finding that NES typically outperforms. (Suggested by *AnonReviewer4*)

3. We have added predictive disagreement for the two ensembles with varying vs. fixed architectures in Section 3.2, showing higher diversity in an ensemble with varying architectures. (Suggested by *AnonReviewer4*)

4. We have updated our related work section with suggestions from the reviewers.

---

> ### Author Response · Authors · 2020-11-23
> **Reminder to reviewers before the discussion period ends**
>
> If the reviewers have any questions, generally about the latest version of our paper or our individual responses, we are very happy to answer those before the end of the discussion period. We hope we have addressed most reviewer concerns.

---

### Decision · Program_Chairs · 2021-01-07
**Final Decision**

**Decision:**

Reject

**Comment:**

This paper proposes a new method to perform uncertainty estimation based on ensembles with diverse network architecture.

The reviewers raised a few concerns:
- Although it is ok not to compare with (Tao, 2019), an active analytical comparison with baselines for ensemble diversification should not be overlooked e.g. (Yao et al, 2008), (Olson et al, 2019), (Khurana et al, 2018), etc.
- The approach presented in this paper is not novel in the general idea of searching for or diversifying ensembles
- The reviewers agree that diversity methods can be implemented on top of NES, but it is unclear whether NES+diversity methods would give more over just diversity methods; so either measuring NES+diversity methods or a direct comparison of NES and diversity methods is important.

We encourage the authors address these issues in the next revision.